# Heat of Hydration Analysis and Temperature Field Distribution Study for Super-Long Mass Concrete

Sanling Zhang [1], Peng Liu [1,2,3,4,*], Lei Liu [1,5], Jingxiang Huang [1], Xiang Cheng [1], Ying Chen [1,6], Lei Chen [4], Sasa He [7], Ning Zhang [7] and Zhiwu Yu [1,2,3]

[1] School of Civil Engineering, Central South University, 22 Shaoshan Road, Changsha 410075, China
[2] National Engineering Research Center for High Speed Railway Construction, Changsha 410075, China
[3] China Railway Group Ltd., 69 Fuxing Road, Beijing 100039, China
[4] China Railway No. 10 Engineering Group Co., Ltd., 2000 Shunhua Road, Jinan 250101, China
[5] School of Physical and Technology, Yili Normal University, 448 Jiefang West Road, Yining 835000, China
[6] School of Civil Engineering, Central South University of Forestry and Technology, 498 Shaoshan Road, Changsha 410004, China
[7] Hunan Zhongda Design Institue Co., Ltd., 68 Shaoshan Road, Changsha 410075, China
* Correspondence: liupeng868@csu.edu.cn

**Abstract:** In this study, the combination of ordinary cement concrete (OCC) and shrinkage-compensating concrete (SCC) was utilized to pour super-long mass concrete. The temperature and strain of the concrete were continuously monitored and managed actively after pouring. The investigation focused on the temporal and spatial distribution patterns of the temperature field, the temperature difference between the core and surface, and the strain evolution. Based on the constructed hydration exothermic model of layered poured concrete, the effects of the SCC, molding temperature, and surface heat transfer coefficient on the temperature field were analyzed. The results show that the temperature of super-long mass concrete rises quickly but falls slowly. SCC exhibits higher total hydration heat than OCC. The temperature field is symmetric along the length but asymmetric along the thickness due to varying efficiency of heat dissipation between the upper and lower parts of the concrete. After final setting of the concrete, the strain varies opposite to the temperature and peaks at $-278$ µε. A few short cracks are observed on the end of the upper surface. Moreover, the numerical simulation results are in good agreement with the measured results. Increasing the molding temperature and surface wind speed increases the temperature difference between the core and surface. Conversely, increasing the thickness of the insulation layer is an effective way to curtail this difference. Thermal stress analysis is carried out and shows that lowering the molding temperature of SCC and increasing the thickness of insulation material can effectively reduce thermal stress.

**Keywords:** super-long mass concrete; temperature field; strain; thermal stress; shrinkage-compensating concrete; numerical simulation

## 1. Introduction

Mass concrete is widely used in dams, nuclear power plants, bridge piers, and large foundations. However, effective crack control remains a significant challenge for scholars globally. Maruyama et al. [1] showed that temperature variation and shrinkage are the main factors affecting the cracking of early-age mass concrete. On the one hand, the initial pouring stage is marked by intense hydration heat release, limited thermal conductivity, and rapid surface heat dissipation. These conditions incite pronounced disparities between the core and surface temperatures, resulting in uneven expansion and consequent tensile stress accumulation. When this stress surpasses the ultimate tensile strength of the concrete, temperature-induced cracks ensue. On the other hand, concrete experiences volume reduction, known as shrinkage, driven by water evaporation and hydration reactions during the hardening stage. This shrinkage contributes significantly to surface cracking

and deformation, undermining the structural robustness and long-term durability of mass concrete.

Effective management of early hydration heat is imperative to mitigate the risk of early cracking in mass concrete structures caused by temperature-induced stress. Methods of controlling concrete temperature can generally be divided into two methods: pre-treatment and post-treatment. Pre-treatment methods aim to lower the concrete mix temperature before pouring. For example, Klemczak et al. [2] showed that using aggregates with appropriate thermal properties (i.e., low specific heat, high thermal conductivity, and low coefficient of thermal expansion) reduces the risk of concrete cracking. Yang and Longarini et al. [3,4] demonstrated that incorporating a substantial admixture of supplementary cementitious materials effectively curtails the heat of hydration while enhancing compressive strength and durability. In addition, admixtures that reduce the hydration heat, such as inhibitors of temperature rise [5], phase change materials [6], and special industrial wastes [7,8], can be incorporated into concrete for temperature control. Post-treatment methods focus on controlling the peak temperature and the temperature difference between the core and surface after concrete pouring. Ha et al. [9] developed an automatic curing system that can effectively control the temperature difference between the core and surface and reduce the possibility of cracks. Han [10] considered the layered system (i.e., layered pouring) as a curing scheme, comparing it to an insulating system and pipe cooling. Notably, the layered system significantly reduces peak temperature and maximum temperature difference, offering substantial time and cost economies, particularly when horizontal construction joints are well-handled. Furthermore, despite the merits of pipe cooling, including high cooling efficiency and ample cooling capacity, certain drawbacks persist, such as generating considerable temperature disparity around cooling pipes and rapid temperature decline during the later stages [11–13].

In addition to controlling temperature, managing shrinkage is another pivotal consideration in mitigating cracks in mass concrete. Proper wet-curing methods can effectively limit the development of early shrinkage strain [14]. Furthermore, scholars started to investigate changing the mineral composition of cementitious materials, among which expansive agents are widely studied for compensating concrete shrinkage [15–17]. Presently, there are three main types of expansion agents: sulfoaluminate, CaO, and MgO, of which sulfoaluminate is most widely used. Liu et al. [18] showed that the addition of an appropriate amount of sulfoaluminate expansion agent can effectively increase the expansion strain of concrete and significantly improve the anti-cracking performance of mass underground concrete structures. Zhang et al. [19] investigated the effect of mineral admixture on the expansion performance of sulphoaluminate expansive agents, and the results revealed that adding fly ash and slag reduces the expansion effect of the expansive agent. Additionally, Yan et al. [20] pointed out that at 20 °C, sulfoaluminate expansive agent can impede the hydration reaction of composite cementitious materials. In contrast, at 45 °C, the maximum exothermic rate and total exothermic amount of composite cementitious materials mixed with sulfoaluminate expansive agents were similar to or exceeded those of pure cement.

Most of the experiments mentioned above have been conducted in laboratory settings, with relatively fewer studies conducted on actual engineering sites. Dwairi et al. [21] monitored the concrete temperature during the construction of a high-performance concrete bridge. They showed that the concrete reached a peak temperature of 70 °C after 10 h of pouring. Ouyang et al. [22] realized the real-time temperature measurement of a reservoir intake tower using a distributed temperature sensing system. After solving the temperature field by finite element simulation, the thermal stress was calculated. The cracking rate was then incorporated to enable the precise prediction of concrete cracking risk under various simulated conditions. This improved the efficiency of temperature and crack control for mass concrete and also provided a new platform for intelligent construction and management of mass concrete. Wang et al. [23] assessed three crack control methods (i.e., pipe cooling, induced joint, and alternative bay construction) for a super-long underground diaphragm wall. The results indicated that the alternative bay construction

method was the most effective measure considering temperature and constraint effects. Amin and Chu et al. [24,25] devised a thermal stress device that accurately predicts hydration thermal stress in mass concrete structures. They successfully applied it to a dam at a Cofferdam construction site. In addition, several scholars adopted hydration heat modeling. Huang et al. [26] conducted a parametric analysis for a 1:5 scale pier based on verifying model accuracy, where parameters such as adiabatic temperature coefficient, cement type, environment temperature, and convection coefficient were studied. In addition, the thermal stresses in the model with and without pipeline cooling were analyzed, and the results showed that the maximum thermal principal stress of the model with pipeline cooling was 2.9 MPa, which was substantially reduced compared with the model without pipeline cooling. Xie et al. [27] obtained the cement hydration kinetic model parameters suitable for composite cementitious systems using the back-propagation (BP) neural network algorithm. Employing these parameters, the simulated temperature field of mass slab concrete closely matched the measured data. However, as aforementioned, field tests and finite element analysis investigations in the early-age temperature field of super-long mass concrete remain relatively scarce. Therefore, field tests, temperature field simulation, and thermal stress analysis of super-long mass concrete are of great practical significance for effectively controlling the temperature and reducing crack generation in such structures.

In this study, field tests were conducted on the mass concrete structure of an aircraft ground dynamics test platform, where a layered pouring mode combined with the pouring of SCC at both ends was adopted. Real-time monitoring of temperature and strain in the mass concrete occurred after concrete pouring. Numerical simulation of the hydration temperature field was carried out using the FE program ABAQUS version number 2021. Parametric analysis was undertaken to assess the impact of SCC, molding temperature, and surface heat transfer coefficient on the early-age temperature field of the super-long mass concrete. Moreover, thermal stress analysis was conducted to predict the location and possibility of thermal cracks.

## 2. Experimental Program

### 2.1. Project Background

The project, located in the Kaifu district of Changsha city, is the first ground dynamics test platform for large aircraft in China. The main structure of the platform is made of concrete, with a length of 1067 m and dimensions exceeding 1 m in width and height. Therefore, the construction is carried out according to the super-long mass concrete. To avoid the shrinkage cracks caused by the excessive length of the mass concrete, the method of segmental pouring combined with post-pouring expansive strengthening bands was employed [28]. The schematic and on-site photo of this super-long mass concrete are depicted in Figures 1a and 1c, respectively. The poured segments hold dimensions of 28,000 mm in length, 7500 mm in width, and 1450 mm in height. Focusing on segment A, temperature and strain fields were subject to monitoring, and the schematic and on-site photo of segment A are displayed in Figures 1b and 1d, respectively. Additionally, the central region of segment A employs OCC, while SCC is utilized at both ends.

### 2.2. Test Methods

The concrete was pumped from the concrete mixing plant and delivered to the construction site by concrete mixer trucks. The concrete was poured in three successive layers with heights of 500 mm, 500 mm, and 450 mm, respectively. The whole pouring process was continuous and orderly, with all the secondary layers of concrete being poured before the first layer of concrete was initially set. After pouring, the surface was plastered and shaped, and a plastic film was spread after plastering in order to prevent a large number of penetrating air holes from the hydration in the structure. In addition, a moist geotextile was used to insulate and moisturize the exposed surface of the structure after the concrete set. This curing measure effectively controls the temperature of the concrete surface, prevents the occurrence of dry cracking phenomenon, and improves the strength and durability of the concrete.

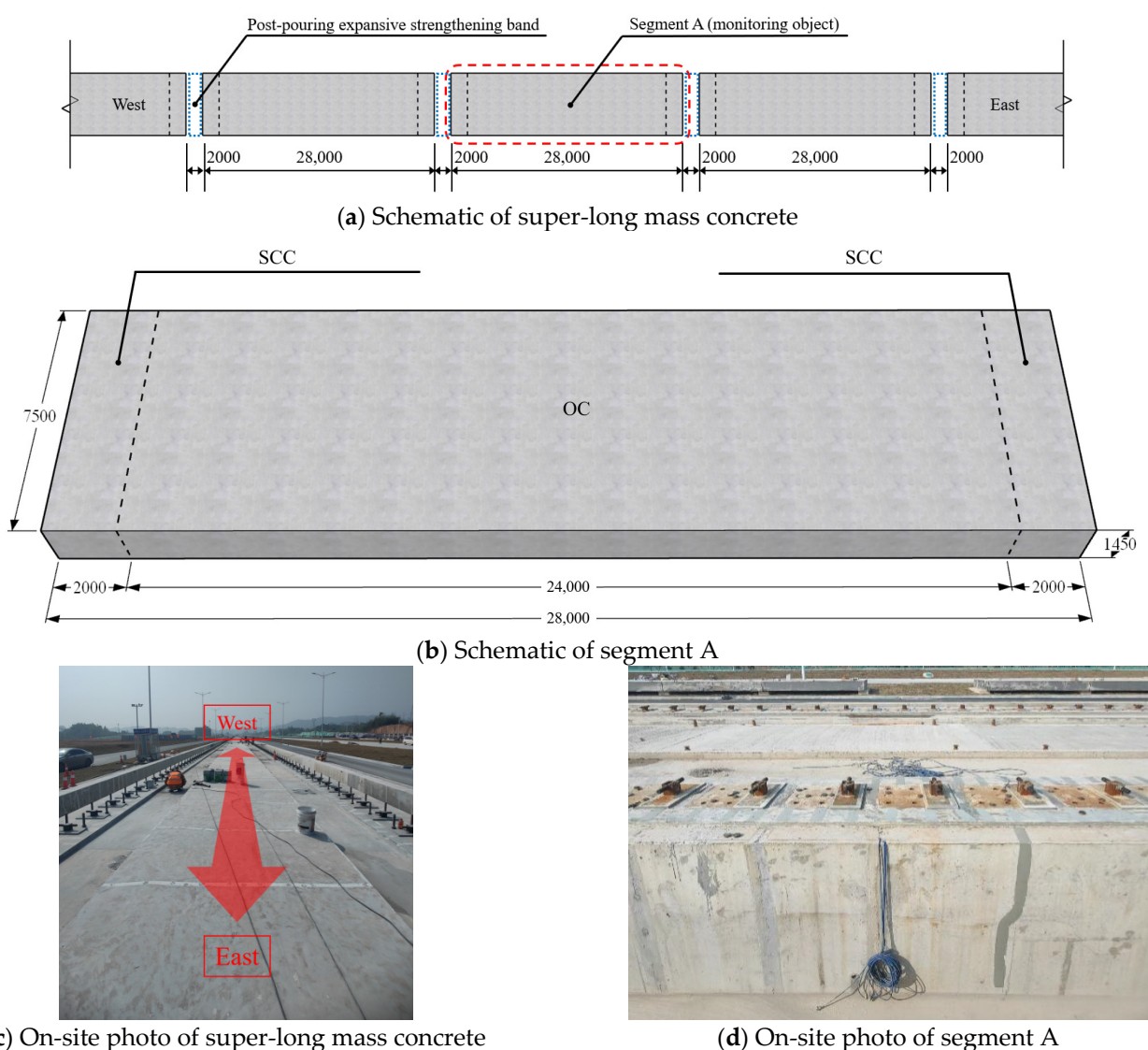

(**a**) Schematic of super-long mass concrete

(**b**) Schematic of segment A

(**c**) On-site photo of super-long mass concrete

(**d**) On-site photo of segment A

**Figure 1.** Schematics and on-site photos of the project (unit: mm).

To prevent the emergence of temperature cracks due to the significant temperature difference between the core and surface, as well as the emergence of shrinkage cracks due to the internal shrinkage of the concrete, the temperature and strain are measured and controlled from the beginning of the concrete pouring. During the monitoring period, the data collection frequency is less than 4 h each time, and the collection can be stopped when the difference between the concrete surface location at 40–100 mm and the environment temperature is less than 20 °C, which all meet the requirements of the standard GB 50666-2011 [29]. Moreover, the molding temperatures of OCC and SCC were 23 °C and 26 °C, respectively, in accordance with the requirement of GB 50496-2018 [30].

### 2.3. Materials and Mix Proportion of Concrete

The mixing proportions of OCC and SCC are listed in Table 1. The water-to-binder ratios (W/B) of OCC and SCC were 0.35 and 0.33, respectively. The P·O 42.5 Portland cement was provided by Shuangfeng Hailuo Cement Co., Ltd. (Loudi, China). River sands with a fineness modulus of 2.7 were used as fine aggregates, and crushed stone with continuous grading from 5 to 25 mm was used as coarse aggregate. Grade II fly ash and S95 slag were added to improve concrete properties. In addition, polycarboxylate superplasticizers with a water-reducing rate of 30% and sulphoaluminate expansive agents were utilized.

**Table 1.** Mixing proportions of concrete (kg/m$^3$).

| Materials | W/B | Cement | Water | Fine Aggregate | Coarse Aggregate | Fly Ash | Slag | Water Reducer | Expansive Agent |
|---|---|---|---|---|---|---|---|---|---|
| OCC | 0.35 | 290 | 150 | 738 | 1070 | 80 | 60 | 12.5 | - |
| SCC | 0.33 | 280 | 150 | 737 | 1050 | 75 | 50 | 13.1 | 45 |

*2.4. Mechanical and Shrinkage Properties of Concrete*

According to GB/T 50081-2019 [31], an electro-hydraulic servo material machine was used to measure the cubic compressive strength of concrete with dimensions of 100 mm × 100 mm × 100 mm, as shown in Figure 2a. Concrete prismatic specimens with dimensions of 100 mm × 100 mm × 515 mm were utilized to determine the drying shrinkage using a horizontal concrete shrinkage tester, as shown in Figure 2b. The test steps were performed in accordance with GB/T 50082-2009 [32]. After the curing was completed and the specimen was moved to a constant temperature and humidity environment. the initial length of the specimens was measured. The temperature and relative humidity were maintained at (20 ± 2) °C and (60 ± 5)%, respectively. The measured results are shown in Figure 2c and reveal that the SCC with an expansive agent can significantly reduce shrinkage compared to the OCC without an expansive agent. Furthermore, the measured cubic compressive strength and drying shrinkage ratio of OCC and SCC at 7 d and 28 d are presented in Table 2.

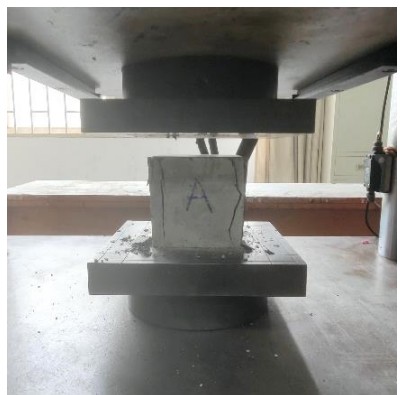

(**a**) Mechanical properties test of concrete

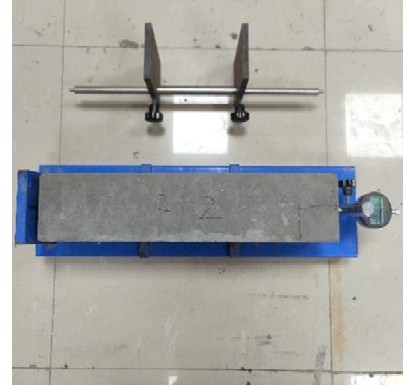

(**b**) Drying shrinkage test of concrete

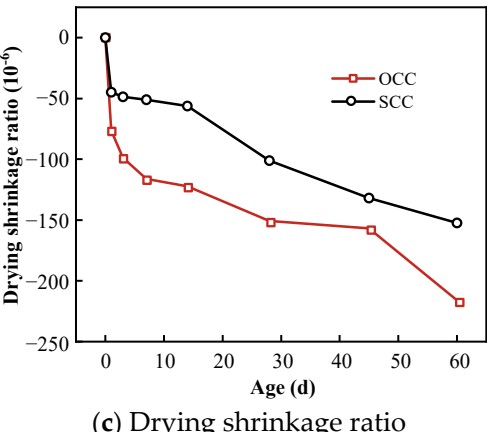

(**c**) Drying shrinkage ratio

**Figure 2.** Mechanical properties and shrinkage test of concrete.

**Table 2.** Mechanical properties and drying shrinkage ratio of concrete.

| Concrete | $f_{cu}$ (MPa) | | Drying Shrinkage Ratio ($10^{-6}$) | |
|---|---|---|---|---|
| | 7 d | 28 d | 7 d | 28 d |
| OCC | 32.7 | 48.7 | −116.1 | −150.8 |
| SCC | 33.2 | 49.2 | −51.1 | −101.2 |

Note: $f_{cu}$ is the cubic compressive strength of concrete with a dimension of 100 mm × 100 mm × 100 mm.

### 2.5. Test Instruments

Temperature sensors and vibrating wire strain gauges (VWSG) were adopted for this mass concrete temperature and strain monitoring, and the data were collected by the integrated intelligent reading instrument, as shown in Figure 3. The VWSG consists of a steel string, coil, thermistor, and protective tube. It is a temperature self-compensating VWSG, with a strain range of ±1500 με and a sensitivity of 1 με. The temperature sensor with an accuracy of ±0.5 °C has a measuring range of −40 °C to 125 °C. All the parameters of these instruments are in accordance with the standard GB/T 51028-2015 [33].

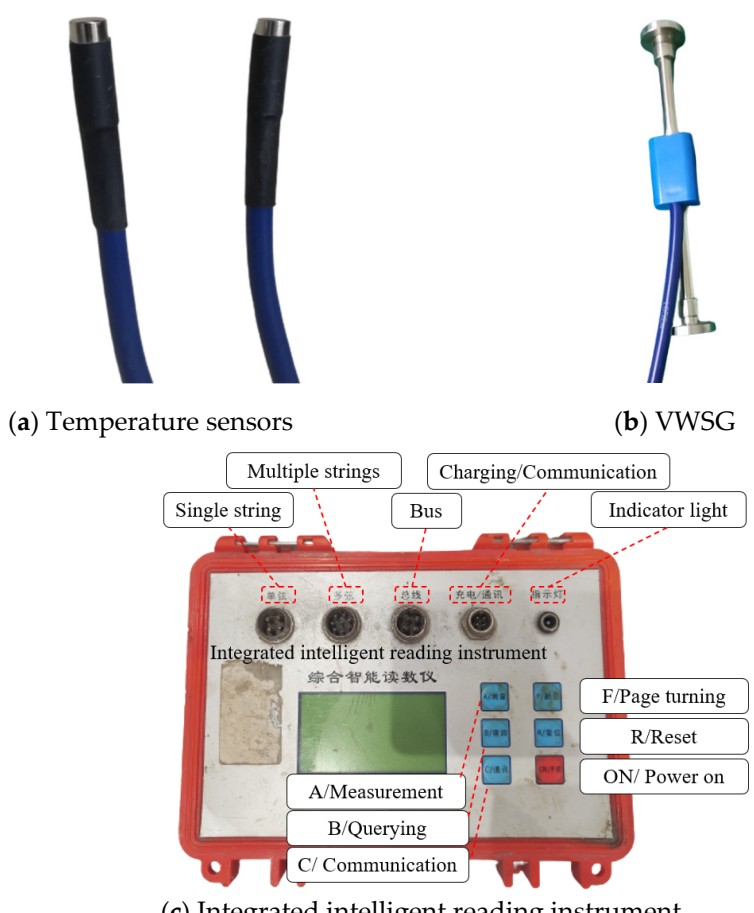

(**a**) Temperature sensors        (**b**) VWSG

(**c**) Integrated intelligent reading instrument

**Figure 3.** Monitoring instruments of temperature and strain.

### 2.6. Measurement Points Layout

The layout of measurement positions and points in the test is shown in Figure 4. Four monitoring sections (i.e., A, B, C, and D) were established along the longitudinal axis. The distances from the sections to the eastern end were 1 m, 7 m, 14 m, and 27 m, respectively. Each section was equipped with three measurement positions labeled 1, 2, and 3, situated at distances of 100 mm, 1875 mm, and 3750 mm, respectively, from the inner surface of the southern formwork. Furthermore, to monitor the temperature variation of the concrete inside the formwork, an additional position (i.e., C0) was established in proximity

to the formwork in section C. Three measurement points (i.e., 1, 2, and 3) were vertically arranged at each measurement position for the A, B, and D sections. These measurement points were located at distances of 100 mm, 725 mm, and 1350 mm, respectively, from the bottom surface of the concrete foundation. To investigate the effect of the bottom concrete foundation on the heat dissipation after the hydration exotherm of mass concrete, measurement point 0 was set at the location where section C was attached to the upper surface of the concrete foundation. Meanwhile, to study the temperature variation of the surface concrete considering the combined effects of internal hydration heat and external environment temperature, measurement point 4 was positioned 50 mm below the upper surface of the concrete at the measurement position C3. Taking A1-2 as an example to explain the designation: A represents section A; 1 indicates that the measurement position is 1; 2 indicates that the measurement point is 2.

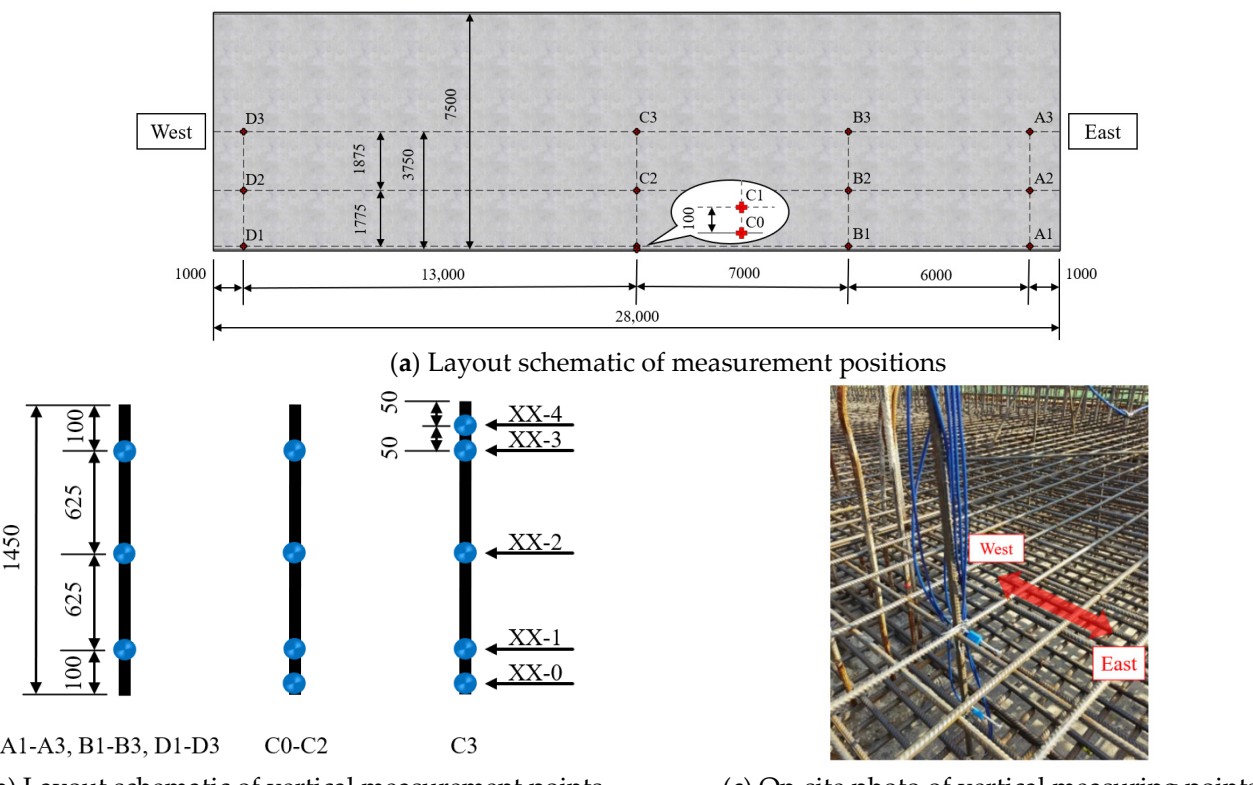

(**a**) Layout schematic of measurement positions

(**b**) Layout schematic of vertical measurement points

(**c**) On-site photo of vertical measuring points

**Figure 4.** Layout of measurement positions and points (unit: mm).

All measurement points were equipped for temperature sensing, whereas strain measurements were performed at nine designated points (i.e., C1-1 to C1-3, C2-1 to C2-3, and C3-1 to C3-3). Meanwhile, all VWSG were oriented along the length direction since the main strain of super-long mass concrete is along the length direction. In addition, a separate temperature sensor was installed on-site to record the environment temperature.

## 3. Results and Discussion

### 3.1. Distribution of Temperature Field

The concrete pouring commenced on the first day at 11:00 a.m. and was completed by 3:00 a.m. the following day, totaling a pouring of 16 h. Temperature and strain monitoring began at the beginning of pouring and ended at 2:00 p.m. on day 8 for a monitoring period of 171 h. In the early monitoring period, the weather was mainly sunny, the highest temperature reached was 32.5 °C, and the temperature difference between day and night was large. Subsequently, cold and rainy weather dominated the third to fifth days, and the temperature dropped significantly compared with the previous two days. In the last

three days, the environment temperature rose slightly and fluctuated. The environment temperature variations for the eight-day monitoring period are depicted in Figure 5.

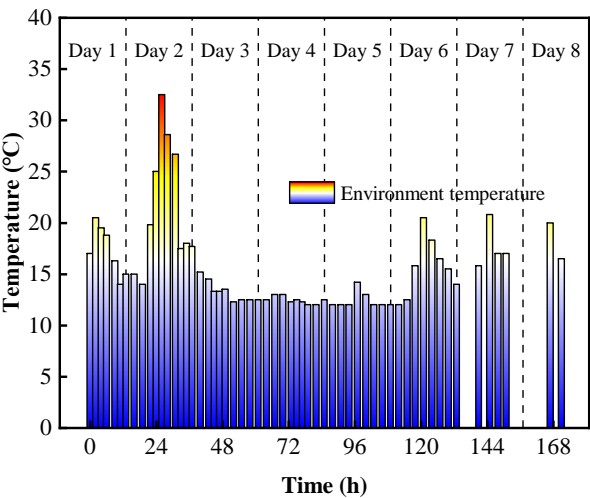

**Figure 5.** Environment temperature of the monitoring period.

Sections A and D are symmetrical along the length of the mass concrete with respect to the mid-span section (i.e., section C). The temperature time-varying curves at measuring point 2 of the two sections are shown in Figure 6. Except for the slight difference in cooling rates between the measuring points A1-2 and D1-2 near the formwork, the curves of the other symmetrical measuring points basically coincide, indicating that the temperature field of mass concrete in section A is symmetrically distributed along the mid-span section. Consequently, in the subsequent analysis of experimental findings, the temperature variations across sections A, B, and C can be considered as representative of the variations in the internal temperature field of the whole mass concrete.

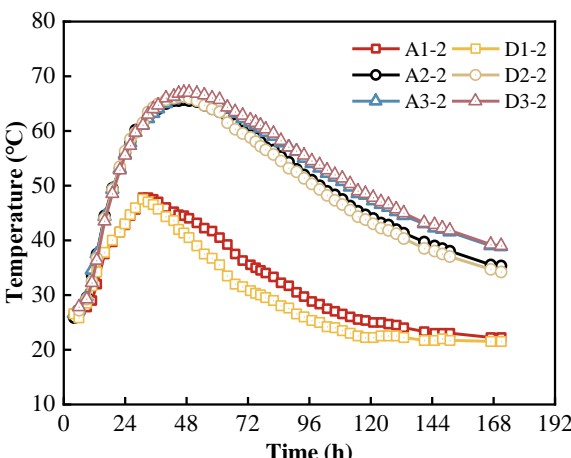

**Figure 6.** Comparison of temperature variation in sections A and D.

### 3.1.1. Vertical Distribution of Temperature Field

To study the temperature field distribution of mass concrete along the thickness direction, four measurement positions, i.e., A1, A3, B2, and C3, were selected to analyze the temperature variation pattern of vertical measuring points, and the temperature time-varying curves were drawn, as shown in Figure 7.

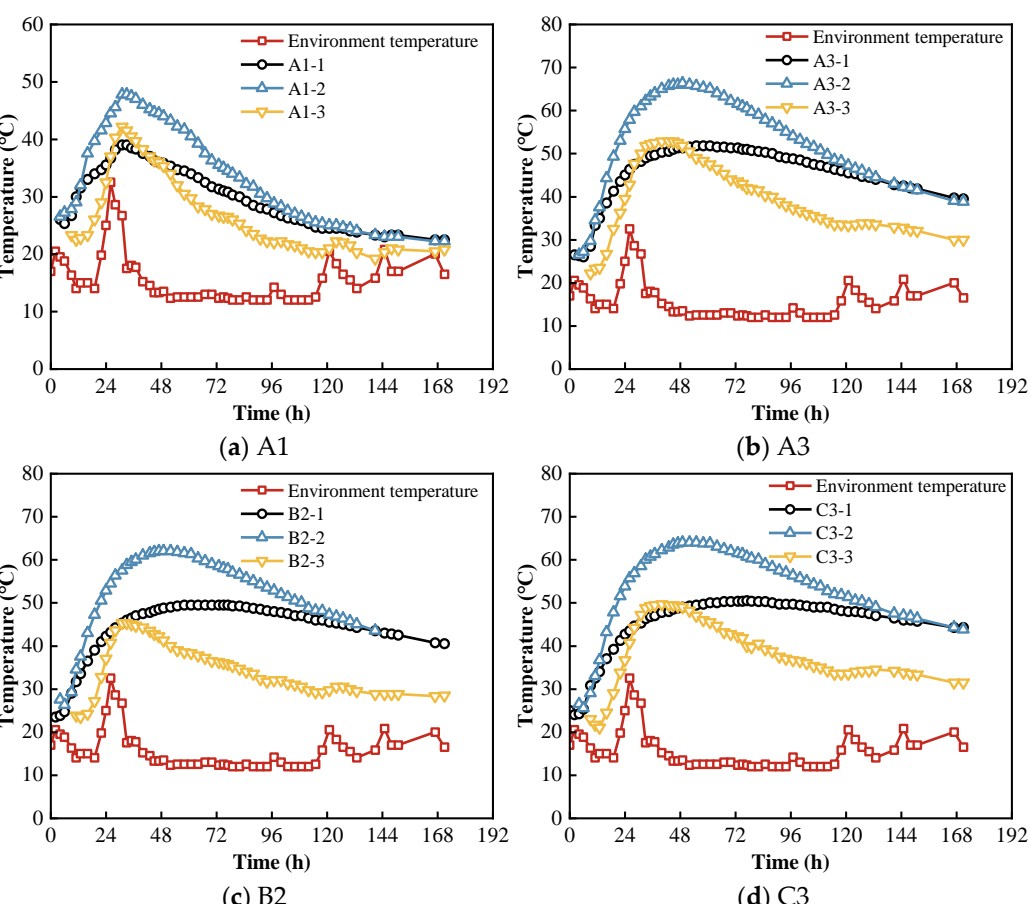

**Figure 7.** Vertical distribution of temperature field.

As seen from Figure 7, the highest temperature at each measurement position occurs at measurement point 2 because measurement point 2 is farther away from the upper surface and the bottom foundation, and the effect of heat dissipation is relatively unfavorable compared to the remaining measurement points. Measurement points 2 and 3 exhibited similar warming and cooling rates and were significantly higher than those of measurement point 1. Except for measurement position A1 near the edge, the center and bottom measurement points of the remaining positions had a significant high-temperature constant stage. Moreover, the impact of environment temperature on the temperature of the concrete surface exhibited an approximate time lag of 6 h.

In summary, the temperature field of early-age mass concrete is mainly affected by the heat of hydration, except for the temperature of surface concrete, which is affected by the environmental temperature change during the cooling stage. Moreover, the maximum temperature was observed at the middle measurement point, the minimum warming and cooling rates were observed at the bottom measurement point, and symmetry was not found in the vertical distribution of the temperature field of mass concrete.

### 3.1.2. Horizontal Distribution of Temperature Field

To investigate the distribution of the temperature field along the length and width of mass concrete, the temperature variation law of measuring points at different height levels of measurement position 3 and section C was analyzed, and the temperature time-varying curves are shown in Figure 8. Furthermore, the maximum temperature and the time taken to reach the maximum temperature of each measurement point at measurement position 3 are summarized in Table 3.

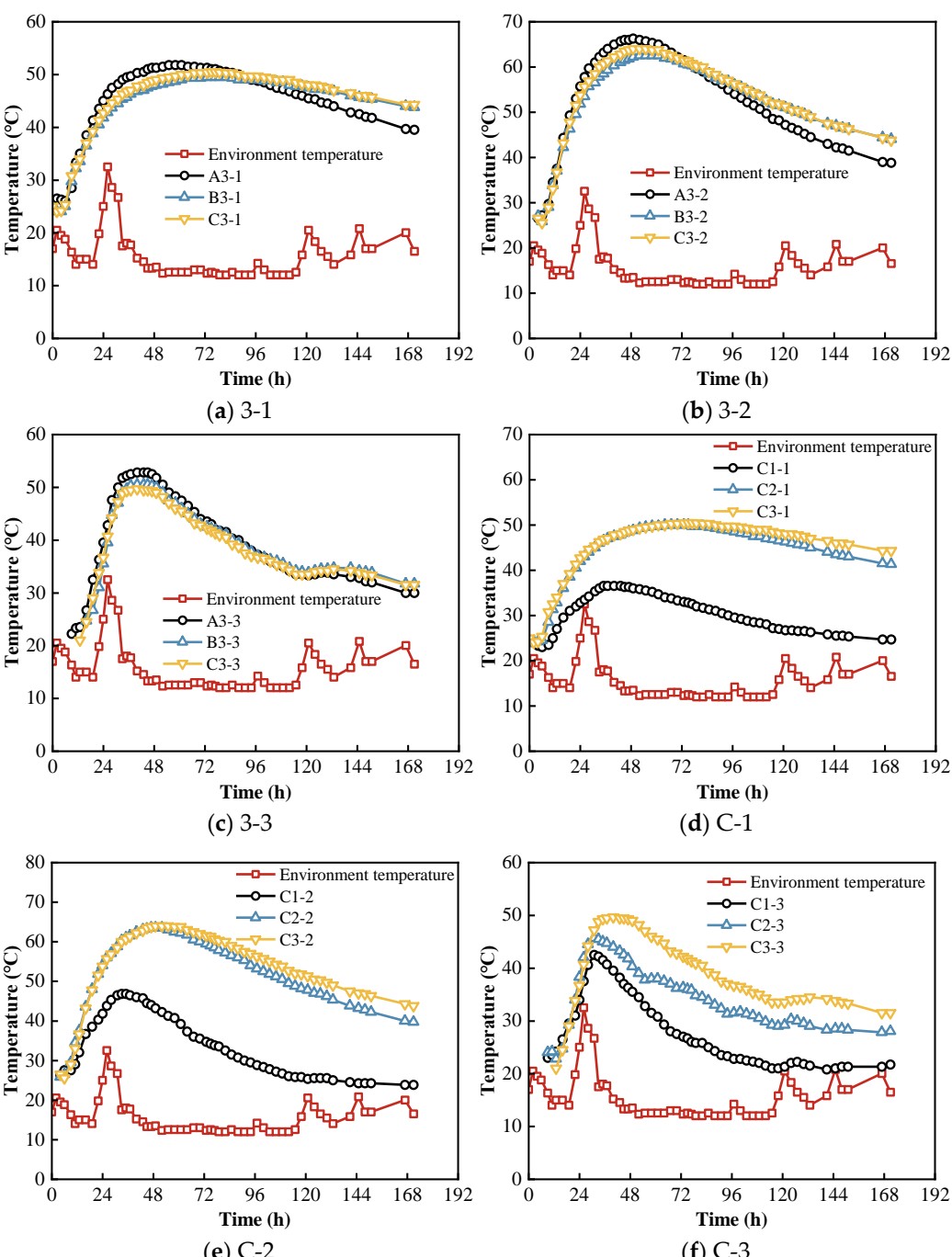

**Figure 8.** Horizontal distribution of temperature field.

**Table 3.** The peak temperature of measurement points and the time taken to reach the peak temperature.

| Measuring Points | A3-1 | B3-1 | C3-1 | A3-2 | B3-2 | C3-2 | A3-3 | B3-3 | C3-3 |
|---|---|---|---|---|---|---|---|---|---|
| $T_P$ (°C) | 51.8 | 49.5 | 50.5 | 66.3 | 62.5 | 64.0 | 52.8 | 50.5 | 49.7 |
| $t_P$ (h) | 55 | 73 | 77 | 49 | 52 | 49 | 40 | 40 | 40 |

Note: $T_P$ is the peak temperature of measurement points; $t_P$ is the time taken to reach the peak temperature.

Figure 8a–c and Table 3 illustrate that the temperature variation trends at different height levels for measurement positions A3, B3, and C3 exhibit a fundamental consistency. However, the highest temperatures along the length direction are observed at measurement

position A3 rather than at the centrally located C3 as anticipated. Furthermore, the central and lower-level measurement points of position A3 attain the peak temperature slightly earlier, around 5 h–10 h ahead, compared to positions B3 and C3 at the same height. This phenomenon is attributed to the utilization of SSC at both ends of the super-long mass concrete structure. Moreover, SCC with higher molding temperature, lower water-to-binder ratio, and an added expansion agent can accelerate the early hydration process compared with OCC. From Figure 8d,e, it is deduced that measurement points located at distances of 3750 mm and 1875 mm from the steel formwork along the width direction exhibit differential temperature variation during the cooling stage because the cooling rate of the measurement points closer to the formwork is slightly faster than the other measurement points. However, the measurement point situated 100 mm away displays a notable temperature variation discrepancy from the other two points at the same height, owing to rapid heat dissipation. Figure 8f shows a gradual decrease in the highest surface temperatures along the width direction from the center to the edges. Notably, C1-3 experiences a slightly higher cooling rate than C2-3 and C3-3, which is because C1-3 can dissipate heat from the template side and the upper layer, while the other two measurement points dissipate heat mainly from the upper layer.

In conclusion, applying SCC containing a high hydration heat expansive agent at both ends of the super-long mass concrete structure results in a shift in the location of the peak temperature occurrence. As a result, the highest temperatures did not occur in the center of the original forecast.

### 3.1.3. Temperature Difference between Core and Surface

The temperature difference between C3-4 and C3-2 is adopted as the temperature difference between the core and surface. The time-varying curve of the temperature difference is shown in Figure 9.

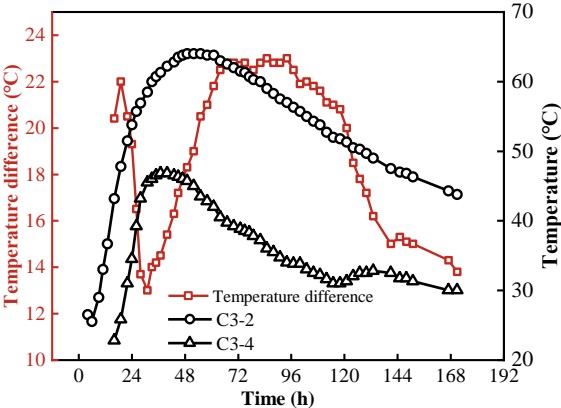

**Figure 9.** Temperature difference between core and surface.

Figure 9 shows that a significant fluctuation is observed in the time-varying curves of the temperature difference between the core and surface. The difference is greater than 20 °C at 16–22 h and 55–118 h. The maximum value of 23 °C of the difference occurs when the age is 85 h. According to GB/T 51028-2015 [33], when the thickness of concrete is less than 1.5 m, the temperature difference between the core and surface should be controlled within 20 °C. Therefore, there is a possibility of cracks due to temperature differences. After the surface concrete is poured, the curve rises for a short period, which is because the newly poured surface concrete has not been fully exothermic, while the poured concrete has entered the stage of fast temperature rise, resulting in a significant temperature difference between the core and surface. As the surface poured concrete begins to become fully exothermic, the temperature rise rate of the surface concrete is greater than that of the internal concrete, causing a decrease in the temperature difference. When the surface concrete temperature rising slows down because of rapid heat dissipation, the temperature

tends to peak, but the internal concrete temperature is still growing, leading to a rise in the temperature difference. When the temperature of the internal concrete peaks, the surface concrete is in the cooling stage, thus the temperature difference between the core and surface reaches a high value. Subsequently, the internal concrete also enters the cooling stage, but owing to that, the cooling rates of the internal concrete and the surface concrete remain identical, the temperature difference is always maintained at a high level in this stage. Until the environment temperature warms up later in the cooling stage, allowing the temperature of the surface concrete to rise, ultimately leading to a decrease in the temperature difference between the core and surface.

From the above results, the following temperature control measures can be suggested: Firstly, long intervals between layers should be avoided when carrying out layered continuous pouring. Reasonable intervals can reduce the temperature difference between the core and surface, reducing the possibility of temperature cracks. Moreover, insulation operation on the surface layer of mass concrete should be conducted at the time when the surface concrete heating slows down and the temperature tends to peak. Additionally, the long cooling stage is a critical stage for the temperature control of mass concrete to prevent cracks, and it is necessary to avoid removing the insulation facilities and formwork too early. Lastly, the temperature difference between the core and surface during the cooling stage is greatly influenced by the environment temperature. If the environment temperature drops sharply during the cooling stage, appropriate thermal insulation measures should be adopted.

### 3.1.4. Effects of Insulation Measures, Concrete Foundations, and Steel Formwork on the Temperature Field

The exothermic hydration and heat dissipation of mass concrete are influenced by factors such as insulation measures, concrete foundations, and formwork in addition to environment temperature. Therefore, to analyze the effects of the above factors on the temperature field of mass concrete, the temperature time-varying curves of some special measurement points were selected for comparison, as shown in Figure 10.

Figure 10a shows that the temperature variation rule of measurement point C3-4, which is only 50 mm away from the upper surface of the concrete, is basically the same as that of measurement point C3-3, which is 100 mm away from the upper surface of the concrete. The temperature difference between C3-4 and C3-3 was always maintained within 4 °C during the whole test stage, and the maximum temperature of the former was 46.8 °C, which was 2.9 °C smaller than that of the latter. The results show that adopting geotextile for surface insulation of mass concrete can effectively slow down the heat dissipation of the concrete surface, thus reducing the temperature difference between the core and surface and the possibility of cracking.

As seen from Figure 10b, compared with measurement point 1 in the same vertical direction, the temperatures at measurement point 0, closely attached to the concrete foundation, decreased, while the temperature changes of both were relatively smooth. This indicates that the heat dissipation of the mass concrete through the bottom concrete foundation is slow, and the existence of the bottom concrete foundation significantly prolongs the duration of the high temperature in the middle and lower layers of the mass concrete.

Figure 10c presents that the four measurement points close to the inner side of the steel formwork reached smaller peak temperatures and maintained a short period near the peak temperature. The difference between the temperatures at these measurement points and the environment temperature is less than 24 °C, and there is an overlapping of temperatures at several temperature measurement points. This is because the steel formwork mainly supports and shapes the concrete, while the thermal insulation is limited.

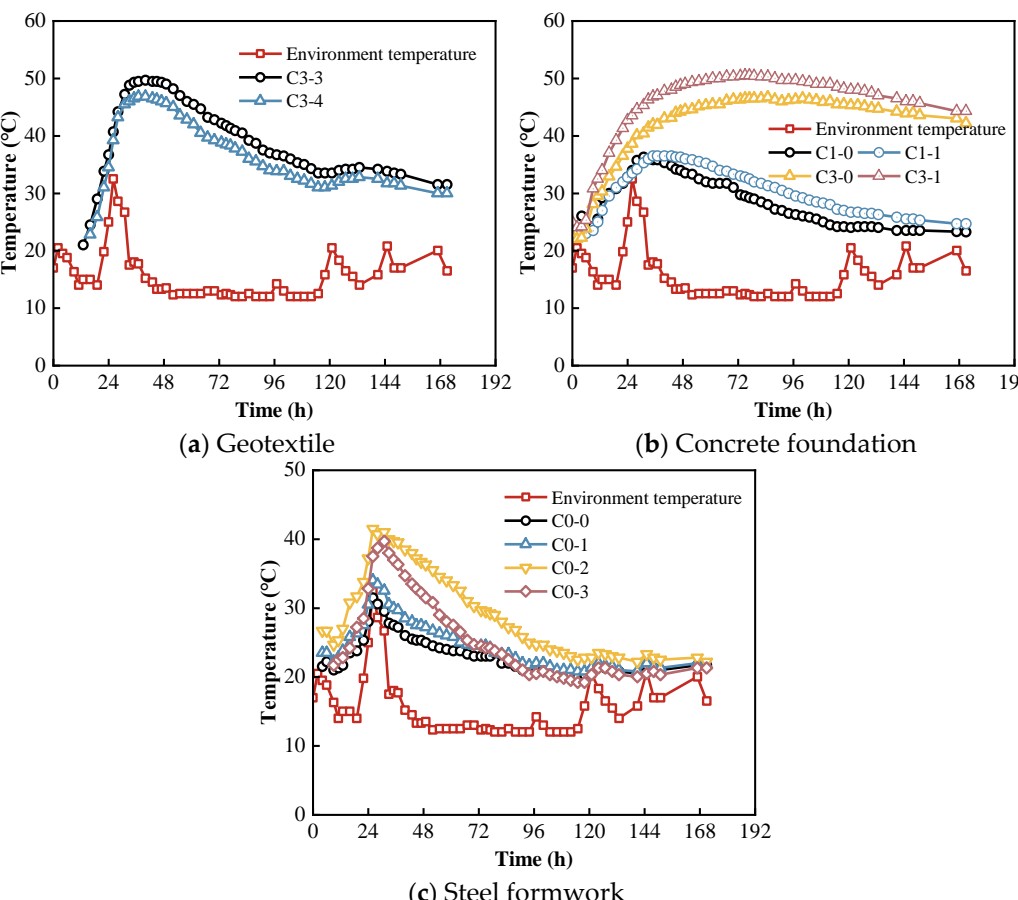

**Figure 10.** Effects of geotextile, concrete foundations, and steel formwork on the temperature field.

*3.2. Strain Field Distribution*

To study the strain variation law of mass concrete at an early age and its relationship with the corresponding temperature, C1 and C3 were selected to show the relationship between strain and temperature, as shown in Figure 11.

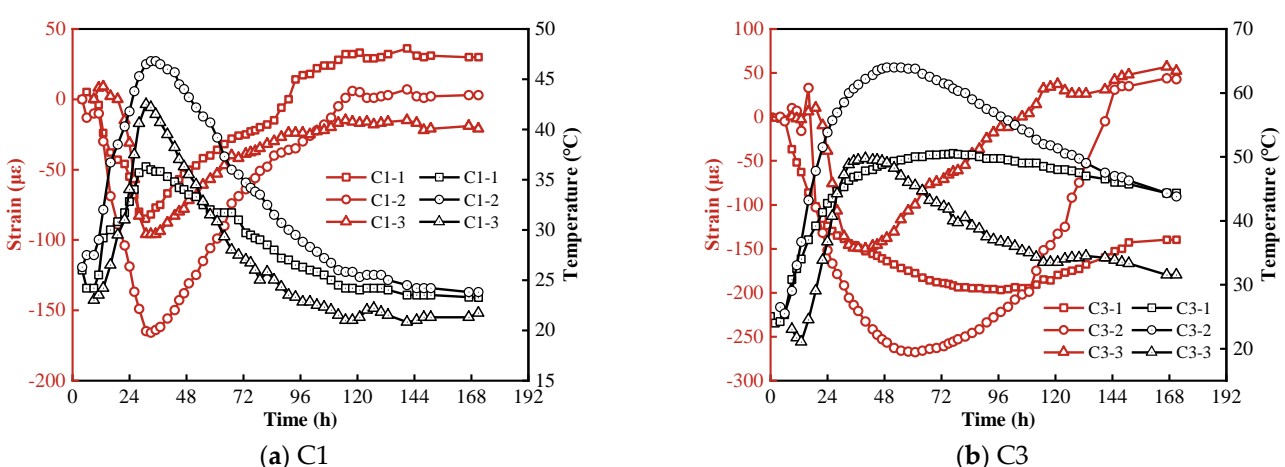

**Figure 11.** Relationship between strain and temperature of concrete.

As indicated in Figure 11, strain exhibits slight fluctuations in the initial 4–6 h after pouring because the pre-setting concrete could not produce sufficient constraints on the VWSG. After the fluctuation stage, the strain starts to change in the opposite direction to the concrete temperature, because the thermal behavior of VWSG is limited by the

surrounding concrete after the concrete final setting. The time taken to reach the turning point is identified as the final setting time of the concrete structure, which coincides with the final setting time (i.e., 480 min) of the concrete provided by the commercial concrete mixing station. Referring to the treatment of strain data by Yeon et al. [34], all strain readings were zeroed to this turning point. The subsequent strain changes are basically opposite to those with temperature changes, with the maximum compressive strain occurring when the peak temperature is reached, which is consistent with the conclusions of the field experiment by Mei et al. [35]. The maximum compressive strains of C1 and C3 occur at C1-2 and C3-2, respectively, with strain values of $-156$ με and $-278$ με. To analyze the relationship between strain and temperature more clearly, the time required to reach the maximum compressive strain and the highest temperature are summarized, as shown in Table 4.

**Table 4.** Peak temperature of measuring points and the time taken to reach the peak temperature.

| Measuring Points | C1-1 | C1-2 | C1-3 | C3-1 | C3-2 | C3-3 |
|---|---|---|---|---|---|---|
| $t_p$ (h) | 31 | 33 | 31 | 77 | 49 | 40 |
| $t_m$ (h) | 31 | 33 | 31 | 97 | 61 | 40 |
| $t_m/t_p$ | 1.00 | 1.00 | 1.00 | 1.26 | 1.24 | 1.00 |

Note: $t_m$ is the time taken to reach the maximum compressive strain.

Table 4 shows that the strain turning points near the side and surface measurement points (C1-1, C1-2, C1-3, and C3-3) occur simultaneously with the temperature turning point. The strain turning points at the internal measurement points (C3-1 and C3-2) occur with a lag of about $25\%t_p$ from the temperature turning point. In addition, during the cooling stage, the strain curves of the concrete surface and side measurement points fluctuate considerably, indicating that the environment significantly affects the strain of mass concrete at an early age.

*3.3. Crack Analysis*

After the completion of the concrete temperature monitoring, crack searching was performed on the mass concrete, and some cracks observed are shown in Figure 12.

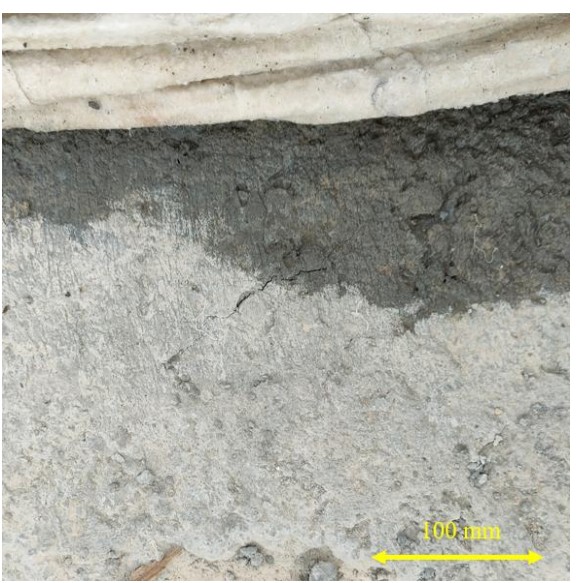

(**a**) Crack 1                                                                                            (**b**) Crack 2

**Figure 12.** Photos of cracks at the test site.

As seen in Figure 12, short cracks ranging from 100 mm to 200 mm in length were observed on the end of the upper surface, and no surface cracks or penetration cracks were

found. There were approximately four such short surface cracks. If these cracks do not expand, they are generally not harmful to the structure. In addition, the reasons for these short surface cracks are as follows: Firstly, the high wind speed at the construction site makes the water evaporation from the concrete surface too fast, which may lead to cracking of the concrete surface. Secondly, the temperature difference between the core and surface may cause thermal stresses, leading to the formation of cracks. Finally, inappropriate curing measures such as poor geotextile cover and untimely rehydration may also lead to the appearance of short surface cracks.

## 4. Numerical Simulation

### 4.1. Heat Conduction Model

The temperature field of mass concrete is affected by the thermal properties of concrete (i.e., thermal conductivity and specific heat), adiabatic temperature rise, surface heat dissipation coefficient, environment temperature, and molding temperature. Due to the conservation of heat, the heat absorbed by an increase in temperature was equal to the sum of the internal heat of hydration, and the net heat inflow from the outside is expressed as follows:

$$c\rho \frac{\partial T}{\partial t} \mathrm{d}t \mathrm{d}x \mathrm{d}y \mathrm{d}z = \left[ Q + \lambda \left( \frac{\partial^2 T}{\partial x^2} + \frac{\partial^2 T}{\partial y^2} + \frac{\partial^2 T}{\partial z^2} \right) \right] \mathrm{d}x \mathrm{d}y \mathrm{d}z \mathrm{d}t \tag{1}$$

where $c$ is the specific heat of concrete (kJ/(kg·°C)); $\rho$ is the density of concrete (kg/m$^3$); $T$ is the temperature of concrete (°C); $t$ is the time (h); $Q$ is the heat generated by hydration heat per unit volume of concrete in unit time (kJ/(m$^3$·h)); $\lambda$ is the thermal conductivity of concrete (kJ/(m·h·°C)).

The above equation is simplified to obtain the heat conduction equation of concrete as follows:

$$\frac{\partial T}{\partial t} = \frac{Q}{c\rho} + \frac{\lambda}{c\rho} \left( \frac{\partial^2 T}{\partial x^2} + \frac{\partial^2 T}{\partial y^2} + \frac{\partial^2 T}{\partial z^2} \right) \tag{2}$$

Under adiabatic conditions, the rate of the temperature rising of concrete after hydration can be expressed as:

$$\frac{\partial \theta}{\partial t} = \frac{Q}{c\rho} \tag{3}$$

where $\theta$ is the adiabatic temperature rise of concrete (°C).

Combining Equations (2) and (3), the heat conduction equation can be rewritten as:

$$\frac{\partial T}{\partial t} = \frac{\partial \theta}{\partial t} + \frac{\lambda}{c\rho} \left( \frac{\partial^2 T}{\partial x^2} + \frac{\partial^2 T}{\partial y^2} + \frac{\partial^2 T}{\partial z^2} \right) \tag{4}$$

The adiabatic temperature rise of concrete can be estimated from the hydration heat of cementitious materials by the following equation:

$$\theta(t) = \frac{q(t)(W + kF)}{c\rho} \tag{5}$$

where $q(t)$ is the cumulative heat of hydration of cement at age $t$ (kJ/kg); $W$ is the amount of cement (kg/m$^3$); $k$ is the discount factor; $F$ is the amount of supplementary cementitious materials (kg/m$^3$).

The empirical equation for the exothermic heat of hydration of cement is adopted as follows [36]:

$$q(t) = q_0 \left( 1 - \mathrm{e}^{-a\left(\frac{t}{24}\right)^b} \right) \tag{6}$$

where $q_0$ is the final heat of hydration (kJ/kg); $a$ and $b$ are constants and are related to the type of cement.

To intuitively determine the speed of hydration exotherm at different ages, Equation (6) is derived to obtain the hydration heat release rate per unit volume of concrete in unit time as follows:

$$q_v = \frac{abq_0}{24} \left( \frac{t}{24} \right)^{b-1} \mathrm{e}^{-a\left(\frac{t}{24}\right)^b} \tag{7}$$

Furthermore, in the calculation of the temperature field, Xie et al. [27] showed that considering the effect of temperature on the hydration exotherm is more accurate than ignoring the effect of temperature. Therefore, to accurately analyze the temperature field evolution of mass concrete, the concept of equivalent age is introduced. The equivalent age equation is used in the following form [37]:

$$t_\mathrm{e} = \sum_0^t \exp\left[ \frac{E_\mathrm{a}}{R} \left( \frac{1}{273 + T_\mathrm{r}} - \frac{1}{273 + T} \right) \right] \Delta t \tag{8}$$

where $t_\mathrm{e}$ is the equivalent age (h); $E_\mathrm{a}$ is the activation energy of concrete (J/mol), 33,500 J/mol; $R$ is the gas constant, 8.314 J/(mol·K); $T_\mathrm{r}$ is the reference temperature, 20 °C; $\Delta t$ is the time interval (h).

The relationship between temperature and time is established through the above equations. In addition, the initial and boundary conditions need to be determined. On the one hand, the initial instantaneous temperature distribution pattern inside the object is considered as the initial condition, i.e., the molding temperature is regarded as the initial condition of newly poured concrete. In this study, the molding temperatures of OCC and SCC are taken as 23 °C and 26 °C, respectively, concerning actual measurements. The initial condition of the concrete foundation is taken as 17 °C, which is equivalent to the initial environment temperature. On the other hand, the temperature interaction law between the concrete surface and the surrounding medium (e.g., air) is taken as the boundary condition, which is represented by the surface heat transfer coefficient.

*4.2. Finite Element Model*

To provide guidance for subsequent engineering applications while optimizing time and economic resources, the FE software ABAQUS with the version number of ABAQUS 2021 was employed to simulate the temperature field of massive concrete at an early age. To verify the accuracy of the FE model, the temperature clouds and temperature time-varying curves obtained from the simulation at different moments were compared with the experimental results. In addition, an extended study was carried out based on the model to analyze the effects of various parameters on the temperature field.

A user-defined subroutine named HETVAL was developed using the FORTRAN programming language to simulate the heat generation process during concrete hydration. The layered pouring was achieved by establishing different analysis steps in the pre-treatment analysis stage. The boundary condition between concrete and air was determined by the surface heat transfer coefficient in the interaction module. The time-varying environment temperature was realized by the amplitude module. Tie constraints were employed to establish connections between concrete and the foundation, among different layers of concrete, and between OCC and SCC. Heat transfer was facilitated by defining thermal conductivity. In addition, considering the influence of the foundation on the temperature change of concrete, the foundation within a depth of 1.4 m below the pouring plane was taken as the simulation object. The size of the foundation model is 30.8 m × 10.3 m ×1.4 m, and fully fixed boundary conditions were defined. Segment A and its corresponding concrete foundation were built using DC3D8 units, with a total number of 18,732, as shown in Figure 13. A three-layer pouring with thicknesses of 0.5 m, 0.5 m, and 0.45 m for the first, second, and third layers was adopted. The pouring time of the first and second layers was 2 h, and the pouring time of the third layer was prolonged to 4 h because of the shaping process. The interval between layers was 4 h, which was less than the initial setting time of 4.5 h provided by the concrete mixing plant and complied with the requirements of

the concrete pouring in the standard GB 50496-2018 [30]. Notably, this timeline aligned precisely with the actual total pouring duration of 16 h.

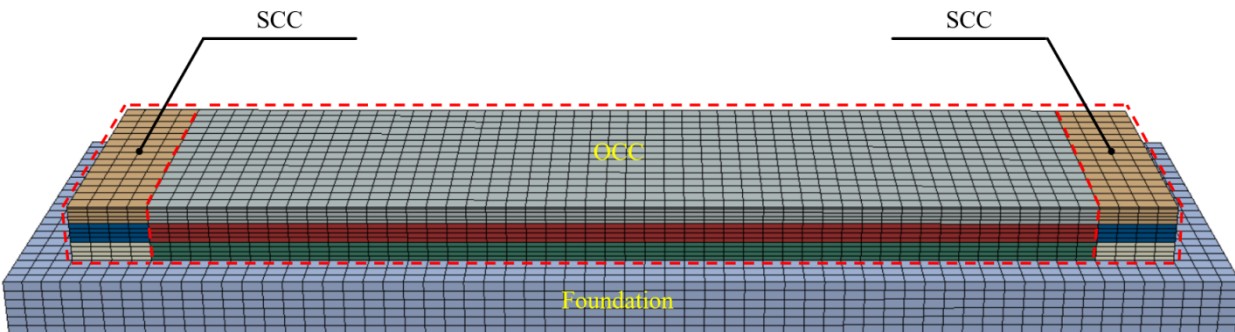

**Figure 13.** Schematic diagram of 3D FE model.

It was essential to input reliable parameters to predict the actual thermal behavior accurately. The density ($\rho$) and thermal conductivity ($\lambda$) of concrete were obtained from previous studies and validated with field data [10,27]. The influence of steel mesh on the thermal conductivity of concrete was considered by specific heat ($c$). The final heat of hydration ($q_0$) and the coefficient of the heat of hydration ($a$ and $b$) were related to the cement type. Referring to the reference [30], the values of $q_0$, $a$, and $b$ were taken to be 330 kJ/kg, 0.69, and 0.56 for P·O 42.5 Portland cement, respectively. Furthermore, conditions such as environment temperature and initial temperature of concrete were used from the actual recorded data. These input parameters to the FE are summarized in Table 5.

**Table 5.** Input parameters of the concrete for the FE model.

| Materials | OCC | SCC | Foundation |
|---|---|---|---|
| $\rho$ (kg/m$^3$) | 2400 | 2400 | 2350 |
| $c$ (kJ/(kg·°C)) | 0.97 | 0.97 | 0.8 |
| $\lambda$ (kJ/(m·h·°C)) | 7.9 | 7.9 | 7.9 |
| Initial temperature (°C) | 23 | 26 | 17 |

*4.3. Model Validation*

The temperature evolution of a quarter-volume mass concrete from the commencement of layered pouring to an age of 171 h is illustrated in Figure 14. The first layer of concrete is poured with OCC and SCC with a height of 0.5 m, and the molding temperatures are 23.0 °C and 26.0 °C, respectively. As the second layer is poured, the first layer of concrete is in the rapid hydration stage, the high-temperature area is concentrated in the middle and lower part of the first layer of concrete, and the maximum temperature reaches 39.6 °C. In contrast, the temperature of the foundation remains relatively stable. With the completion of the third layer pouring, a high-temperature area is observed in the first layer and lower portion of the second layer, and the maximum temperature reaches 56.2 °C. At the age of 48 h, the peak temperature of 71.0 °C is reached, and the high-temperature area is concentrated in the second layer of concrete. Meanwhile, the temperature of the foundation close to the mass concrete rises significantly due to the absorption of heat transferred from the upper concrete. At the age of 171 h, the mass concrete is in the cooling stage, the temperature of the upper edge part decreases significantly, the high-temperature region moves down, and the maximum temperature drops to 51.0 °C.

The time-varying temperature curves of measurement points A3-2, C3-2, and C3-4 are shown in Figure 15. As the age increases, the temperature increases and then decreases, the peak temperature of the central measurement point is larger than that of the surface measurement point, and the cooling rate of the measurement point near the surface is obviously larger than that of the measurement point in the center of the concrete. The temperature trends obtained from simulation and measurement are basically in agreement.

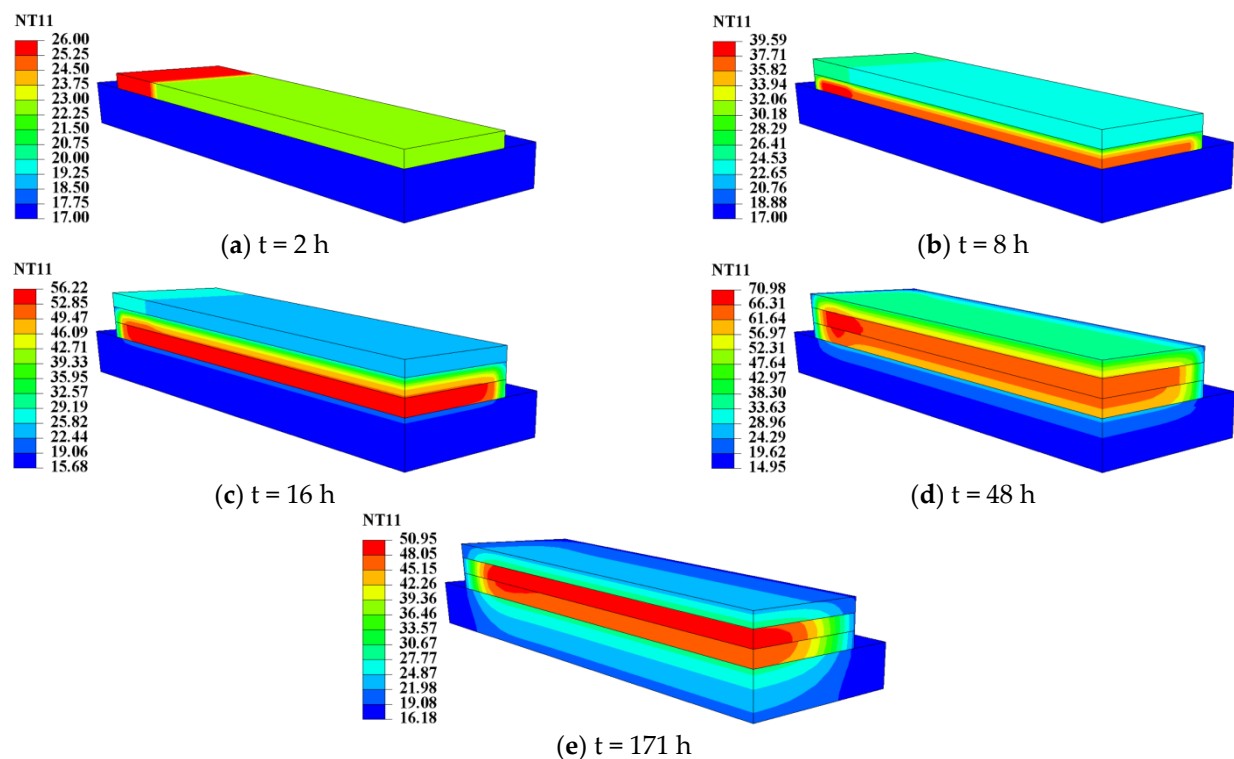

**Figure 14.** Temperature clouds at different times.

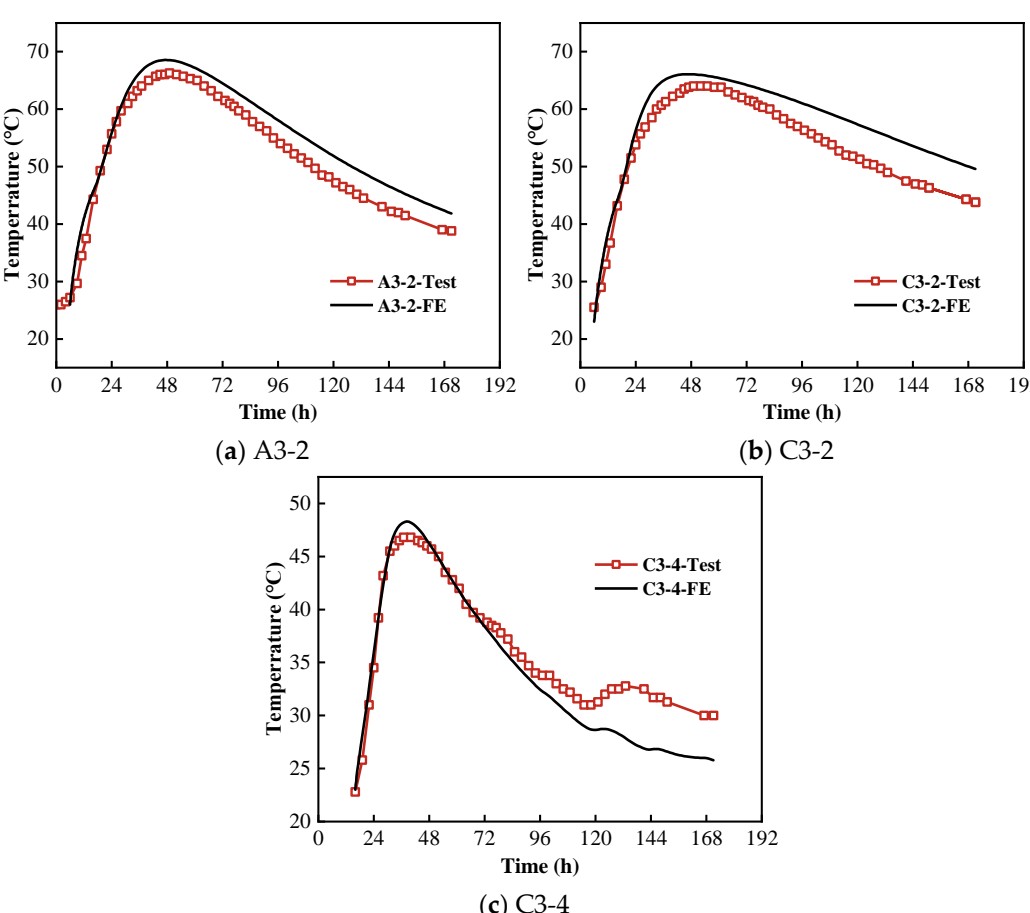

**Figure 15.** Comparisons between the test temperature time-varying curves and FE results.

Furthermore, to compare the simulation results with the measured results more precisely, the peak temperature and the time taken to reach the peak temperature at the three measurement points are summarized in Table 6, which shows that the difference between the results of the FE analysis and the measured results is no more than 4%. Therefore, the FE analysis model is feasible and can be used for subsequent parametric analysis.

**Table 6.** Differences of peak temperature between the test results and FE results.

| Measuring Points | A3-2 | | C3-2 | | C3-4 | |
|---|---|---|---|---|---|---|
| | $T_p$ (°C) | $t_p$ (h) | $T_p$ (°C) | $t_p$ (h) | $T_p$ (°C) | $t_p$ (h) |
| Test | 66.3 | 49 | 64 | 49 | 46.8 | 37 |
| FE | 67.3 | 48.3 | 66.1 | 47.3 | 48.3 | 38.3 |
| FE/Test | 1.02 | 0.99 | 1.03 | 0.97 | 1.03 | 1.04 |

*4.4. Parametric Analysis*

4.4.1. Influence of SCC

As mentioned in Section 3.1.2, pouring SCC at both ends can alter the temperature distribution of super-long mass concrete. Therefore, studying the effect of using SCC in super-long mass concrete structures on their temperature distribution and maximum temperature is necessary. In addition to the actual pouring mode (i.e., SCC were poured at both ends), two other conditions of fully poured OCC and fully poured SCC were simulated, and the temperature cloud diagrams of the three conditions at the age of 48 h were obtained, as shown in Figure 16.

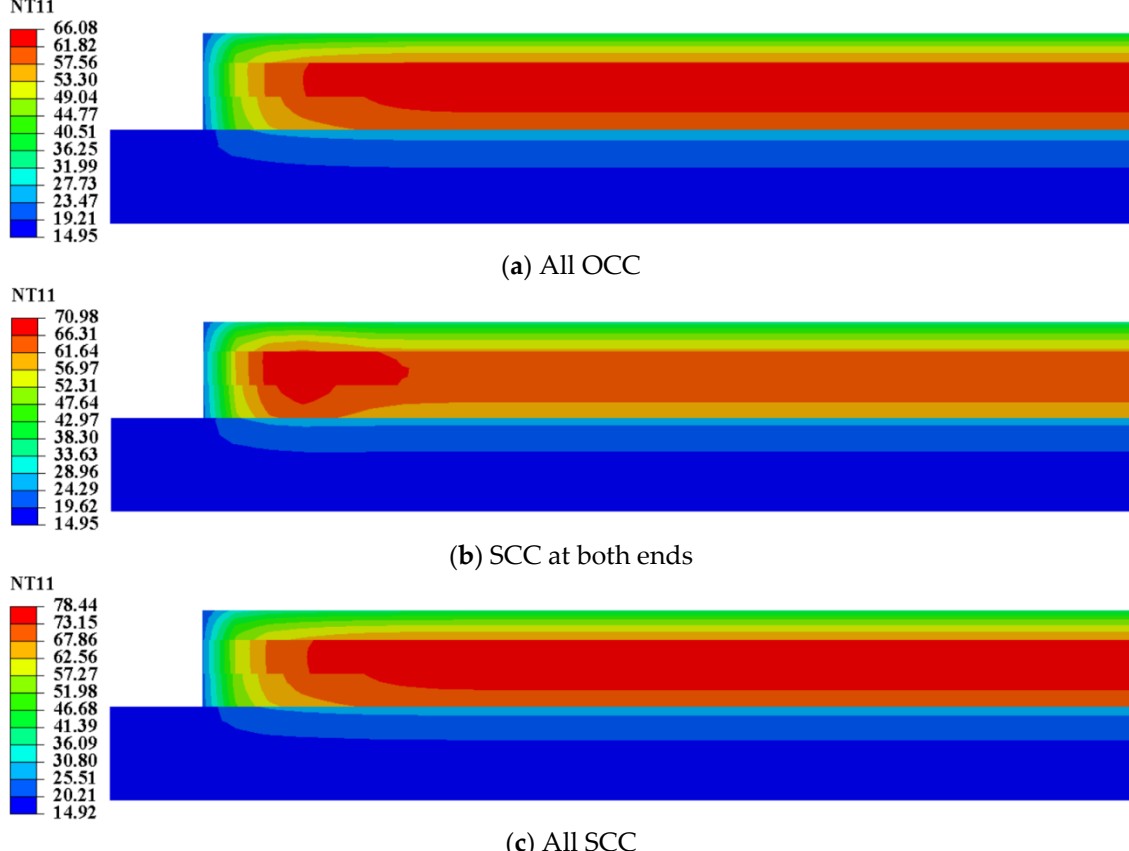

**Figure 16.** Temperature cloud diagrams under different pouring modes.

As seen in Figure 16, the temperature distribution of the mass concrete structure with one type of concrete poured was conventional, i.e., the temperature decreased from inside to outside. In contrast, the temperature distribution of the super-long mass concrete structure with SCC poured at both ends showed a dumbbell shape. The highest temperature occurred near the center of the end-poured SCC instead of the center of the overall structure. The maximum temperatures for the three working conditions were 66.1 °C, 71.0 °C, and 78.4 °C, respectively. Yan et al. [38–40] showed that when the temperature of SCC exceeded 70 °C, the calcite generated by the expansive agent could not exist stably and decomposed. Moreover, delayed ettringite may occur in SCC at temperatures higher than 70 °C. Therefore, the pouring mode of SCC poured at both ends can significantly reduce the maximum temperature compared to all SCC, but the potential for thermal cracking still exists.

### 4.4.2. Influence of Molding Temperature

As the initial temperature of mass concrete, the molding temperature significantly affects the variation and development of its temperature field. Pre-cooled aggregates, cold water mixing, and mixing with ice can be used to reduce the molding temperature in practical projects [41]. In this analysis, the molding temperatures of SCC were taken as 21 °C, 26 °C, and 31 °C, while the molding temperatures of OCC were 18 °C, 23 °C, and 28 °C. The temperature time-varying curves of A3-2, C3-2, and C3-4 in the FE model under different molding temperatures and the temperature difference between C3-2 and C3-4 are shown in Figure 17.

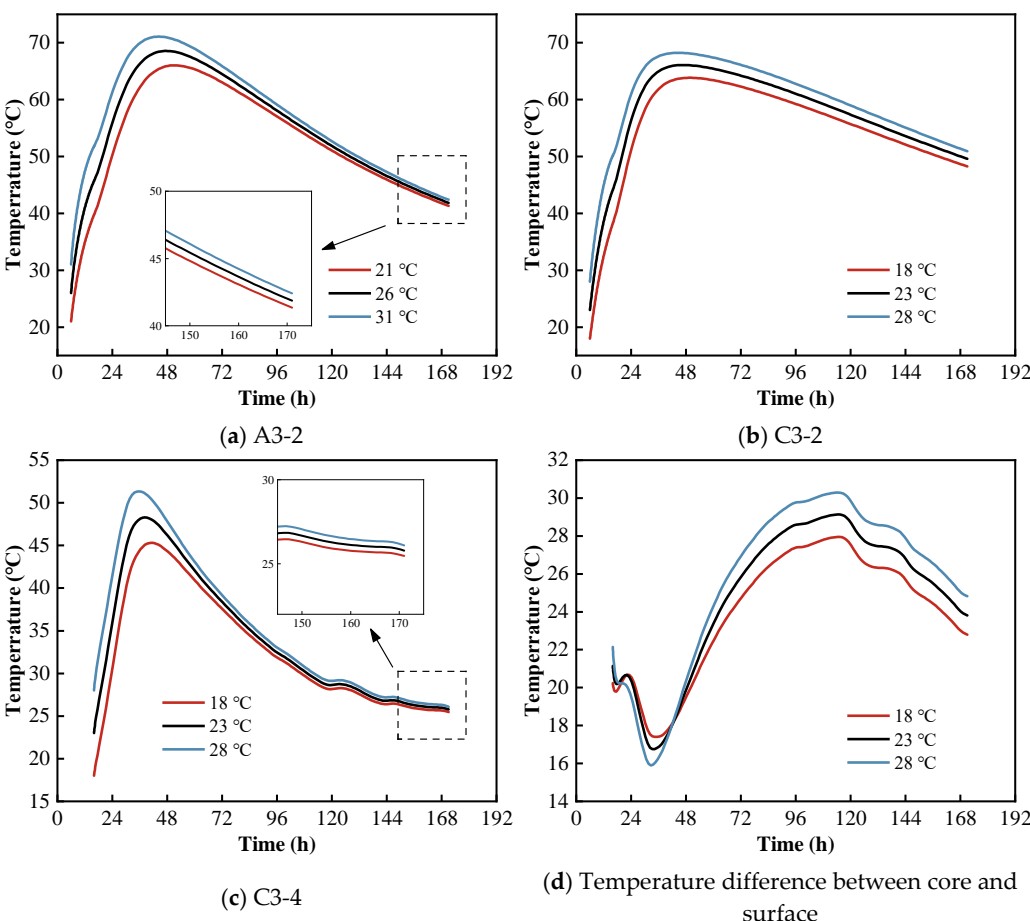

**Figure 17.** Temperature time-varying curves of FE with different molding temperatures.

As seen from Figure 17a–c, the peak temperatures of each measurement point differ significantly under different molding temperatures, but the variation trend is consistent. The specific simulated results are as follows: when the molding temperature of SCC is 21 °C, 26 °C and 31 °C, A3-2 reaches the peak temperatures at 51.3 h, 47.3 h, and 44.3 h, respectively, and the peak temperature is 66.0 °C, 68.6 °C, and 71.1 °C, respectively. When the molding temperature of OCC is 18 °C, 23 °C, and 28 °C, C3-2 reaches the peak temperatures at 49.3 h, 47.3 h, and 44.7 h, respectively, and the peak temperature is 63.8 °C, 66.1 °C, and 68.2 °C, respectively. C3-4 reaches the peak temperatures at 41.3 h, 38.3 h, and 35.7 h, respectively, and the peak temperature is 45.3 °C, 48.3 °C, and 51.3 °C, respectively. As seen from Figure 17d, when the molding temperature is 18 °C, 23 °C, and 28 °C, the corresponding maximum temperature difference between the core and surface is 28.0 °C, 29.1 °C, and 30.3 °C, respectively. The reason for this phenomenon is that increasing the injection temperature increases the cooling rate of the surface concrete, while the cooling rate of the central concrete remains almost constant.

In summary, as the pouring temperature of concrete increases, the peak temperature increases, the time to reach the peak temperature is advanced, and the maximum temperature difference between the core and surface increases. Therefore, the excessively high pouring temperature is not conducive to crack control of mass concrete.

### 4.4.3. Influence of Surface Heat Transfer Coefficient

Most of the cracks initially produced in mass concrete are surface cracks, and some of them develop into penetrating cracks at subsequent stages, seriously affecting the integrity and durability of the structure. Meanwhile, drying shrinkage and temperature stress are the reasons for surface cracks, and the former can be easily solved by moisture curing. The latter is mainly caused by the temperature difference between the core and surface. The internal temperature is determined by the heat of hydration, while the surface temperature is greatly influenced by the external environment. Therefore, it is necessary to investigate the surface insulation effect of mass concrete, defined by the surface heat transfer coefficient in the FE. The surface heat transfer coefficient is mainly related to the type of insulation material, thickness, degree of wetness, and its surrounding wind speed, and the calculation equation is as follows [36]:

$$\beta = 1/(h/(k_1 k_2 \lambda) + 1/\beta_0) \tag{9}$$

where $h$ is the thickness of the insulation layer (mm); $k_1$ is the wind speed correction coefficient, and 1.3 is taken to represent impermeable insulation when the wind speed is less than 4 m/s; $k_2$ is the degree of humidity correction factor, while 1 and 3 are taken to represent the dry and wet materials, respectively; $\lambda$ is the thermal conductivity of the thermal insulation material (kJ/(m·h·°C)); $\beta_0$ is the heat transfer coefficient between the thermal insulation material and surrounding environment (kJ/(m·h·°C)).

$\beta_0$ is mainly related to the wind speed around the concrete surface and is calculated as follows:

$$\beta_0 = 21.06 + 17.58 v^{0.910} \tag{10}$$

where $v$ is the wind speed around the concrete surface (m/s).

In this section, the effects of wind speed around the concrete surface and the thickness of insulation material (i.e., geotextile) on the temperature field are analyzed. The surface heat transfer coefficients of different working conditions are calculated based on $\lambda$ taking 0.188 kJ/(m·h·°C), as shown in Table 7.

**Table 7.** Surface heat transfer coefficient under different working conditions.

| Items | $v$ (m/s) | | | | | $h$ (mm) | | | |
|---|---|---|---|---|---|---|---|---|---|
| | $v = 0$ | $v = 1$ | $v = 2$ | $v = 3$ | $v = 4$ | $h = 0$ | $h = 2$ | $h = 5$ | $h = 8$ |
| $\beta$ (kJ/(m·h·°C)) | 18.4 | 30.6 | 38.9 | 46.3 | 52.7 | 53.0 | 46.3 | 38.9 | 33.6 |

The surface heat transfer coefficients under different working conditions were brought into the model, and the temperature time-varying curves and temperature difference time-varying curves of measuring points C3-2 and C3-4 under different wind speeds were obtained, as shown in Figure 18. As the wind speed increases, the cooling rate of the concrete increases, the maximum temperature of the concrete surface decreases, and the temperature difference between the core and surface increases significantly. In addition, when the wind speed increases from 0 m/s (i.e., absolutely no wind) to 1 m/s, a great change occurs in the surface temperature and the temperature difference between the core and surface, and the change gradually decreases as the wind speed further increases.

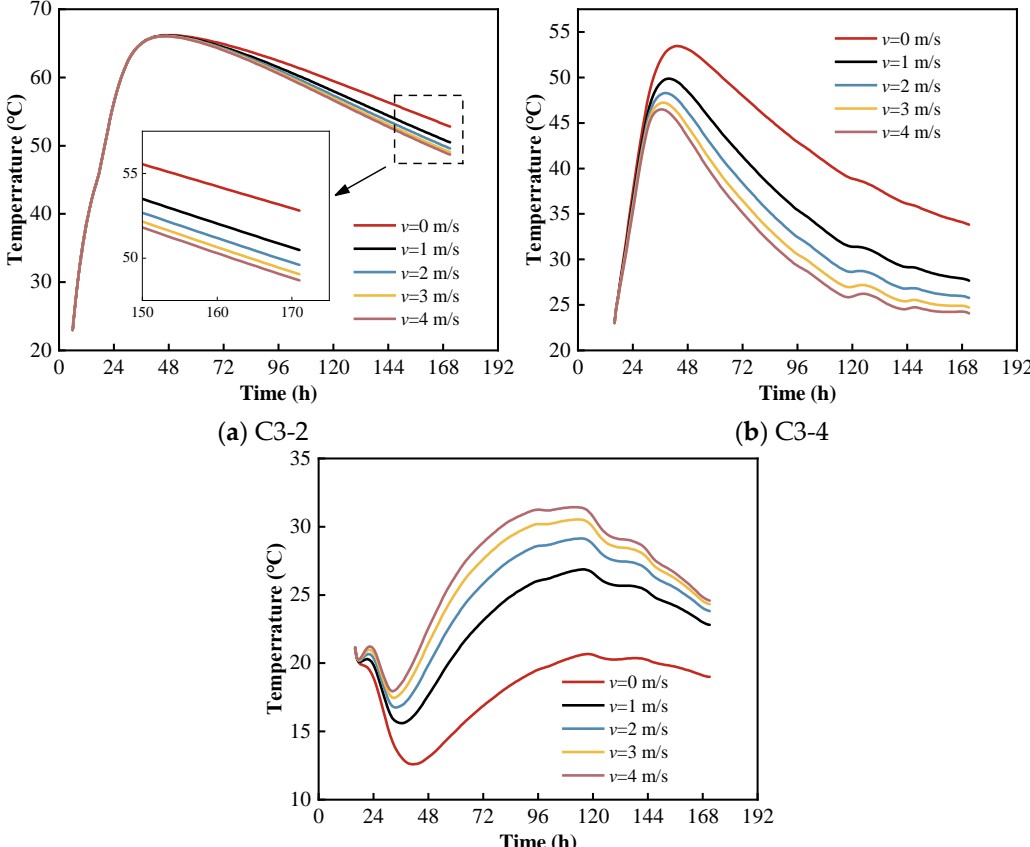

(**a**) C3-2  (**b**) C3-4

(**c**) Temperature difference between core and surface

**Figure 18.** Temperature time-varying curves of FE under different wind speeds on concrete surface.

The temperature time-varying curves of measurement points C3-2 and C3-4 under different thicknesses of the insulation layer and the temperature time-varying curves of the core and surface are shown in Figure 19. With the increase in the thickness of the insulation layer, the temperature of the concrete surface increases, the cooling rate of the concrete decreases, and the temperature difference between the core and surface decreases.

In summary, when the surface wind speed of mass concrete increases, the surface insulation measures should be increased to prevent cracks caused by excessive temperature differences between the core and surface, and increasing the thickness of the insulation layer is an effective method.

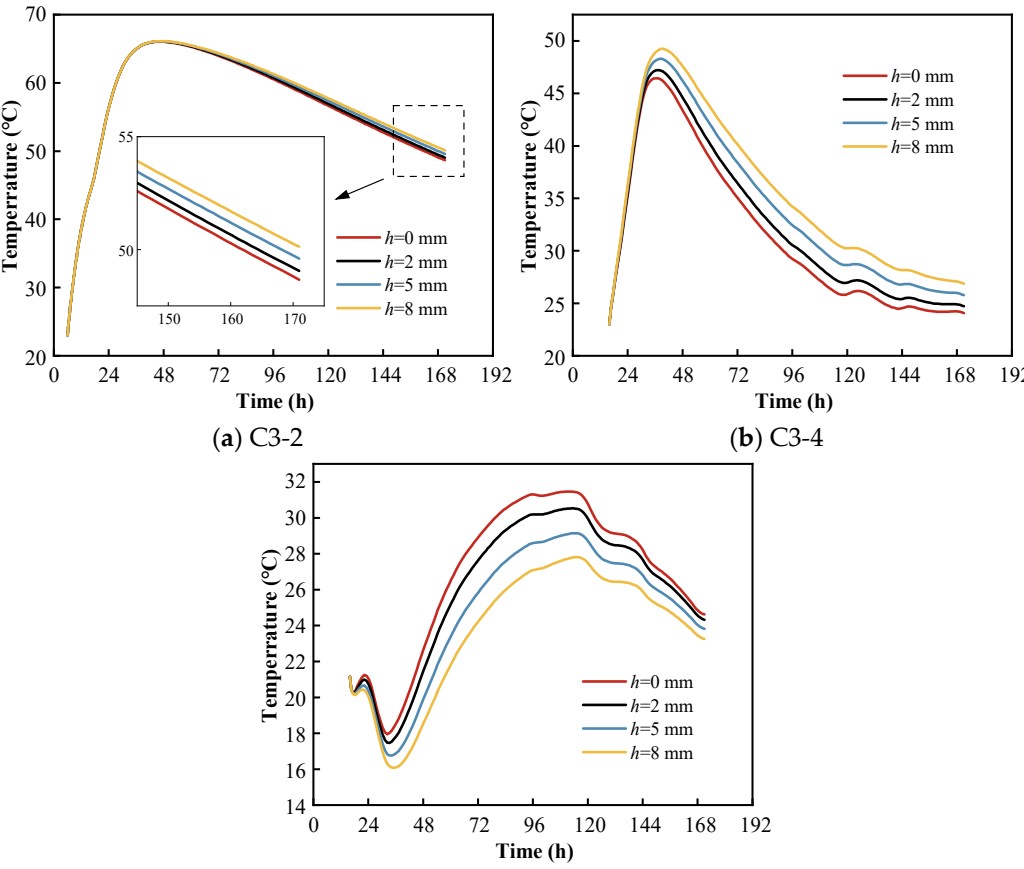

(**a**) C3-2  (**b**) C3-4

(**c**) Temperature difference between core and surface

**Figure 19.** Temperature time-varying curves of FE models with different thicknesses of insulation layer.

## 5. Thermal Stress Analysis

Heat of hydration analysis and temperature prediction play a favorable role in the crack control of mass concrete. To more accurately determine the location and possibility of temperature cracking in super-long mass concrete, thermal stress needs to be predicted and analyzed. The temperature field obtained from the above simulation was imported into ABAQUS, and then the USDFLD subroutine was used to define the field variables, through which the elastic modulus of concrete was correlated with time. The time-varying elastic modulus is referenced from the standard GB 50496-2018 [30] as follows:

$$E(t) = E_0\left(1 - e^{-0.09t/24}\right) \tag{11}$$

where $E(t)$ is the elastic modulus of concrete at time $t$ (MPa); $E_0$ is the elastic modulus of concrete for standard curing for 28 d (MPa), $3.25 \times 10^4$ MPa was adopted to represent the standard value of C40 concrete; $t$ is the time (h).

The crack resistance of concrete can be judged by the following equation:

$$\sigma_{\mathrm{T}}(t) \leq f_{\mathrm{tk}}(t)/K = f_{\mathrm{tk}}\left(1 - e^{-0.3t/24}\right)/K \tag{12}$$

where $\sigma_{\mathrm{T}}(t)$ is the thermal stress of concrete at time $t$ (MPa); $f_{\mathrm{tk}}(t)$ is the tensile strength of concrete at age $t$ (MPa), 2.39 MPa was adopted to represent the standard value of C40 concrete; $K$ is the safety factor for anti-cracking, 1.15.

In addition, the allowable tensile strength (i.e., $f_{\mathrm{tk}}(t)/K$) was calculated separately for the first and third layers of poured concrete at different times and the results are summarized in Table 8.

**Table 8.** Allowable tensile strength of the first and third layers of poured concrete at different times.

| Items | $t = 24$ h | $t = 48$ h | $t = 116$ h | $t = 171$ h |
|---|---|---|---|---|
| First layer | 0.50 | 0.91 | 1.57 | 1.83 |
| Third layer | 0.20 | 0.69 | 1.47 | 1.78 |

Figure 20 illustrates the thermal principal stress clouds of this model at different times. During the temperature rise stage, the concrete expands with heat, resulting in compressive stress in the inner core area and tensile stress on the surface of the structure. At the same time, the tensile stress of the bottom surface is particularly prominent due to the restraint effect of the concrete foundation. The maximum principal tensile stress of 0.42 MPa at this time occurs at the bottom surface corner. When $t = 48$ h, the internal temperature of the concrete peaks and $\sigma_m$ of 0.93 MPa occurs at the bottom edge, slightly exceeding the allowable tensile strength of 0.91 MPa. After entering the cooling stage, the temperature difference between the core and surface gradually increases and reaches the maximum at $t = 116$ h. The maximum principal tensile stress observed during the entire hydration process is 2.32 MPa at the upper surface edge, exceeding the allowable tensile strength of 1.47 MPa by 57.8%, indicating a significant risk of cracking. Notably, the area of temperature stress concentration at this moment is generally consistent with the location of the short cracks found in the actual tests. Finally, as the temperature difference between the core and surface decreases, the maximum principal tensile stress decreases, and its occurrence area converges toward the middle of the upper surface.

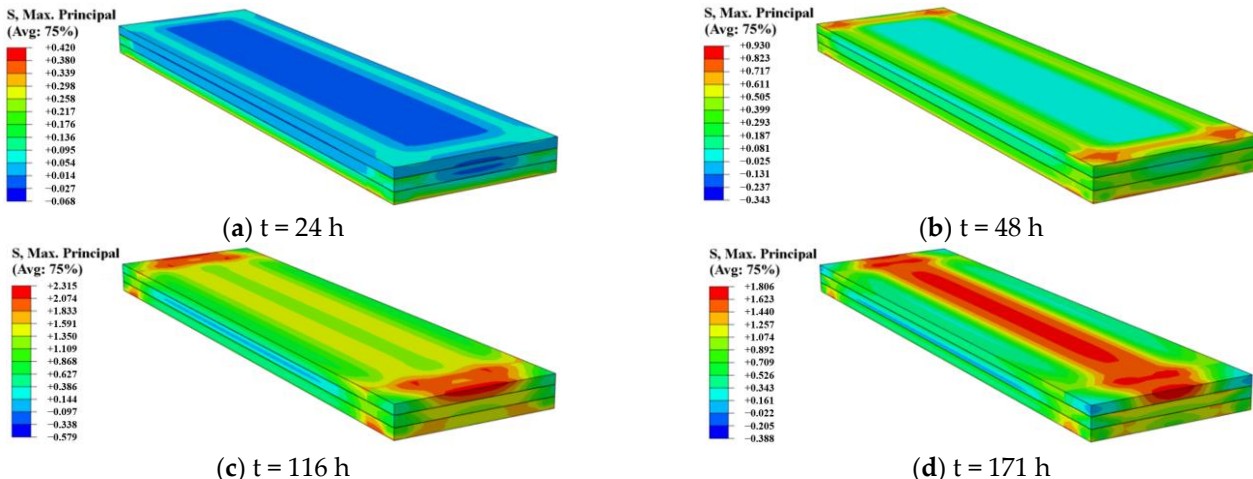

**Figure 20.** Thermal principal stress clouds at different times (unit: MPa).

Further, the effect of molding temperature and thickness of the insulation layer on the thermal stress of super-long mass concrete is investigated and also a reasonable method for reducing thermal stress is explored. Under the two conditions that the molding temperature of SCC is 21 °C and the thickness of the insulation layer is 8 mm, the thermal principal stress clouds at $t = 116$ h are shown in Figure 21.

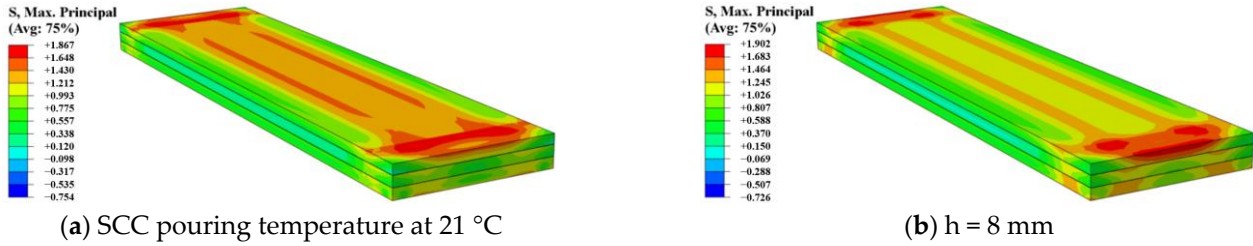

**Figure 21.** Thermal principal stress clouds at t = 116 h under different conditions (unit: MPa).

As shown in Figure 21a, under the condition of SCC pouring temperature at 21 °C, the thermal maximum principal tensile stress at $t$ = 116 h is 1.87 MPa, which is a 19.4% reduction compared to the actual working condition of 2.32 MPa. Meanwhile, the region of concentrated principal tensile stress at this moment exhibits a tendency to converge toward the middle of the upper surface. As seen from Figure 21b, under the condition of a protective layer thickness of 8 mm, the thermal maximum principal tensile stress at $t$ = 116 h is 1.90 MPa, representing an 18.1% decrease compared to 2.32 MPa. This indicates that reducing the molding temperature of SCC and increasing the thickness of the insulation layer both contribute to reducing the maximum principal tensile stress in the structure and mitigating the possibility of thermal cracking.

## 6. Conclusions

Based on the field monitoring and further numerical simulation of the early-age temperature field of super-long mass concrete, the following conclusions can be drawn:

(1) The temperature of super-long mass concrete increases rapidly but decreases slowly. OCC and SCC centers attain peak temperatures of 64.0 °C and 66.3 °C, respectively, after 48 h of pouring. This indicates that pouring SCC at both ends alters the temperature distribution of super-long mass concrete. Additionally, the temperature difference between the core and surface is maintained at a high level for a long period of time during the cooling stage, and a small number of short cracks of 10 mm to 20 mm are generated on the end of the upper surface.

(2) Asymmetry in vertical temperature field distribution arises due to notable variations in heat transfer between the upper air and lower concrete foundation. After the final setting of the concrete, the measured strain, using VWSG, exhibits a contrary trend to temperature changes. The compressive strain increases with the increase in temperature, reaching a maximum value of $-278$ με.

(3) The variation between the simulated temperature results obtained by considering the equivalent age and measured results is not more than 4%. This affirms the precision of the FE analysis and establishes the feasibility of conducting subsequent parametric analysis. The parametric analysis results reveal that the temperature difference between the core and surface increases with rising molding temperature and surface wind speed. Correspondingly, increasing the thickness of the insulation layer is an effective way to reduce the temperature difference.

(4) Thermal stress analysis suggests that the edges and corners of the upper and lower surfaces of the super-long mass concrete are the areas with the maximum thermal principal tensile stresses. Significantly, these stresses reach their maximum when the temperature difference between the core and surface is at its peak, occurring at the end of the upper surface. This observation aligns with the locations of short cracks observed on-site during construction. Additionally, reducing the molding temperature of SCC and increasing the thickness of the insulation layer are both effective in decreasing the maximum principal tensile stress on the upper surface of the structure. This contributes to the mitigation of temperature-induced crack formation.

In summary, the edge locations of super-long mass concrete are more prone to thermal cracking. Therefore, further studies should focus on the selection of concrete composition and the curing measures after concrete placement in these weak locations in order to better solve the thermal cracking problem of super-long mass concrete.

**Author Contributions:** Conceptualization, P.L.; Writing—original draft, S.Z.; Writing—review & editing, P.L., L.L., J.H., X.C., Y.C., L.C., S.H. and N.Z.; Funding acquisition, P.L. and Z.Y. All authors have read and agreed to the published version of the manuscript.

**Funding:** This study was funded by the National Natural Science Foundation of China (grant numbers 52178182, 52108262, and U1934217), Science and Technology Research and Development Program Project of China Railway Group Limited (Major special project, No.: 2020-Special-02, 2021-Special-08, 2022-Special-09; Major project, No.: 2021-Special-02; Key project, No.: 2021-Key-11, No.: 2022-Key-46). Authors also have received research grants from the Natural Science Foundation for Distinguished Young Scholars of Hunan Province (2022JJ10075), the Natural Science Foundation of Hunan Province of China (2020JJ5982), and the Hunan Science and Technology Plan Project (2023SK2014).

**Institutional Review Board Statement:** Not applicable.

**Informed Consent Statement:** Not applicable.

**Data Availability Statement:** Data are contained within the article.

**Conflicts of Interest:** Author Peng Liu was employed by the company China Railway Group Ltd. and China Railway No. 10 Engineering Group Co., Ltd. Author Lei Chen was employed by the company China Railway Group Ltd. Authors Sasa He and Ning Zhang were employed by the company Hunan Zhongda Design Institue Co., Ltd. Author Zhiwu Yu was employed by the company China Railway Group Ltd. The remaining authors declare that the research was conducted in the absence of any commercial or financial relationships that could be construed as a potential conflict of interest.

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
