# Peer review of "Heat of Hydration Analysis and Temperature Field Distribution Study for Super-Long Mass Concrete"

_coatings, doi:10.3390/coatings14030369_

Round 1

Reviewer 1 Report

Comments and Suggestions for Authors

Dear Authors,
thank you for your paper focused on the heat of hydration analysis and temperature field distribution for super-long mass concrete members. The paper presents the field monitoring (in-situ measurements on real structure) and numerical simulations and compares the achieved values between measurements and simulations. The paper is well-written, and comprehensible, individual chapters follow each other correctly and are appropriately supplemented with figures, tables and formulas. Even so, I have comments about the paper.
My comments are:
- line 110, use "MPa" instead of "Mpa",
- fig. 1 - aren't images trimmed from the right side? Are the images displayed in full?
- pages 5-7, table 1 - nothing is displayed, it does not display any texts, it is not clear whether it is a file error or a bad tab.
- figs. 7, 8, 10, 15 - images are trimmed from the right side, the whole images are not displayed, they need to be corrected,
- fig. 12, in the title fig. is "Crack diagrams ...", but two images are displayed, not diagrams - either you need to change images to diagrams, or you need to change the name of the fig to "Photos of cracks of the test site",
- page 29, here is not table 7 (second times), it is table 8, renumber it,
- why is there double numbering of cited articles in references? (eg 1. [1] I. Maruyama ...).
Best regards.

Reviewer 2 Report

Comments and Suggestions for Authors

This study investigates the heat of hydration and temperature distribution in super-long mass concrete, employing a combination of ordinary cement concrete (OCC) and shrinkage-compensating concrete (SCC). The research focuses on understanding the temporal and spatial temperature patterns, strain evolution, and thermal stress analysis. Key findings include the rapid temperature rise and slow fall of super-long mass concrete, the higher hydration heat of SCC compared to OCC, and the symmetric temperature field along the length but asymmetric along the thickness of the concrete. The study suggests various measures for temperature control, including adjusting molding temperature and surface wind speed, and utilizing insulation layers effectively. Additionally, numerical simulations align well with measured results, enhancing the study's credibility.

The introduction provides a comprehensive overview of the challenges associated with crack control in mass concrete structures, emphasizing the significance of effective temperature management. It outlines pre-treatment and post-treatment methods for controlling concrete temperature and managing shrinkage, referencing previous studies for context.

The experimental program details the project background, test methods, materials, mix proportions, mechanical, and shrinkage properties of concrete. It illustrates the layout of measurement positions and points, providing a clear understanding of the experimental setup.

Results and discussion sections delve into the distribution of temperature fields vertically and horizontally, analyzing temperature differences between the core and surface of the concrete. The findings suggest potential temperature control measures, highlighting the importance of proper insulation and timing in the cooling stage.

Overall, the study offers valuable insights into temperature management in super-long mass concrete structures, contributing to the field of concrete construction and crack prevention. The integration of experimental and numerical approaches strengthens the validity of the findings, making it a noteworthy contribution to the literature.

Correction suggestions:

1. In the paragraph starting with "In this study...", consider rephrasing "continuously monitored and actively managed" to "continuously monitored and managed actively."

2. In the same paragraph, consider rephrasing "heat dissipation efficiency" to "efficiency of heat dissipation."

3. In the paragraph starting with "Effective management...", consider rephrasing "attributed to temperature-induced stress" to "caused by temperature-induced stress."

4. In the paragraph starting with "Methods of controlling...", consider rephrasing "admixtures that reduce the hydration heat, such as temperature rising inhibitors, phase change materials, and special industrial wastes" to "admixtures that reduce hydration heat, such as inhibitors of temperature rise, phase change materials, and special industrial wastes."

5. In the paragraph starting with "Proper wet-curing methods...", consider rephrasing "as compensating for concrete shrinkage" to "for compensating for concrete shrinkage."

6. In the paragraph starting with "Nowadays, there are three main types...", consider rephrasing "widely studied as compensating for concrete shrinkage" to "widely studied for compensating concrete shrinkage."

7. In the paragraph starting with "Most experiments above...", consider rephrasing "with relatively fewer studies performed" to "with relatively fewer studies conducted."

8. In the paragraph starting with "To avoid the emergence...", consider rephrasing "and the method of segmental pouring combined with post-pouring expansive strengthening bands was adopted" to "and the method of segmental pouring combined with post-pouring expansive strengthening bands was employed."

9. In the paragraph starting with "The mixing proportions of OCC...", consider rephrasing "with a fineness modulus of 2.7 were used as fine aggregate" to "with a fineness modulus of 2.7 were used as fine aggregates."

10. In the paragraph starting with "According to GB/T 50081-2019...", consider rephrasing "cubic compressive strength of concrete with a dimension" to "cubic compressive strength of concrete specimens with dimensions."

11. In the paragraph starting with "All measurement points were equipped...", consider rephrasing "whereas strain measurements were performed" to "while strain measurements were performed."

12. In the paragraph starting with "Sections A and D are symmetrical...", consider rephrasing "respect to the mid-span section" to "with respect to the mid-span section."

13. In the paragraph starting with "To study the temperature field distribution...", consider rephrasing "temperature variation rule" to "temperature variation pattern."

14. In the paragraph starting with "As seen from Fig. 7...", consider rephrasing "except for the measurement position A1 near the edge" to "except for measurement position A1 near the edge."

15. In the paragraph starting with "In conclusion, applying SCC containing a high hydration...", consider rephrasing "results in a shift in the location of peak temperature occurrence" to "results in a shift in the location of the peak temperature occurrence."

16. In the paragraph starting with "Fig. 9 shows that a significant fluctuation...", consider rephrasing "there is a possibility of temperature difference cracks" to "there is a possibility of cracks due to temperature differences."

17. In the same paragraph, consider rephrasing "until the warming of the environment temperature later in the cooling stage, which allowed the temperature of the surface concrete to rise, eventually leading to a decrease in the temperature difference of core and surface" to "until the environment temperature warms up later in the cooling stage, allowing the temperature of the surface concrete to rise, ultimately leading to a decrease in the temperature difference between the core and surface."

Comments on the Quality of English Language

Overall, the English quality is good. The text is well-structured and technical, with proper use of terminology. However, there are a few areas where phrasing could be improved for clarity and conciseness. Additionally, some sentences could benefit from minor grammatical adjustments. With a few corrections and refinements, the quality of the English could be further enhanced.

Some suggestions for improving the English:

1. Conciseness: Some sentences could be made more concise by removing redundant phrases or unnecessary words

2. Clarity: Clarify any ambiguous or unclear statements to ensure that the meaning is easily understood by the reader.

3. Grammar: Check for grammatical errors such as subject-verb agreement, tense consistency, and punctuation.

4. Word Choice: Use precise and appropriate language to convey ideas effectively.

5. Consistency: Maintain consistency in terminology and writing style throughout the text.

Reviewer 3 Report

Comments and Suggestions for Authors

The paper “Heat of Hydration Analysis and Temperature Field Distribution Study for Super-Long Mass Concrete” reports an interesting research work about the evaluation of concrete behaviour, in terms of heat hydration analysis and temperature field distribution for Super-Long Mass Concrete. In particular, the authors have performed both laboratory tests and numerical analyses. The results obtained have been discussed in detail in Sections 3 and 4. Also the approach used for the execution of the different type of analyses is clearly described in the text. For these reasons, it is opinion of this reviewer that the manuscript can be considered for publication in Coatings after the following minor corrections/improvements:

- in the introduction consider as reported in 10.1007/978-3-031-37123-3_21 and N. Longarini, P, Crespi, M. Zucca, N. Giordano, G. Silvestro. “The advantages of fly ash use in concrete structures”. Inzynieria Mineralna 2014, 15(2), pp. 141 – 145.

- line 110: replace “Mpa” with “MPa”.

- Check the format of the Figures caption.

- Table 1 is not readable in this version of the manuscript.

- Improve the quality of the Figure 3c.

- Add the u.m. of the Figure 4a.

- Check the format of the Figures 7, 8, 10, 15, 17

- Section 4.3: (i) Better described the numerical model shown in Figure 13 (boundary conditions, number of elements, etc..); (ii) Has a construction stage analysis been carried out?: (iii) What are the limitations of the numerical approach used?

- Conclusions: add some considerations concerning the further developments of the research work.

Comments on the Quality of English Language

The paper appears well-written.

Reviewer 4 Report

Comments and Suggestions for Authors

A fairly comprehensive article presents the results of research into heat of hydration and the temperature field for very long concrete. This topic is interesting not only from a theoretical point of view, but is of considerable importance for numerous applications in the construction industry.

After a detailed study of the article, I can state that its content is well processed and brings a number of new findings in the given field of research.

Sensors adequate for the given purpose in this field of technology were used for the experimental work, the results of the measurements are graphically presented and compared with the results of the computational model.

The use of the ABAQUS program supplemented with a own subroutine in the FORTRAN language appears to be suitable for solving this task, it would be interesting for the reader to add in ch. 4.2 at least basic information about the finite element used.

However, I have a larger number of comments on the current formal editing of the article. Some descriptions of the Figues "disintegrated" (eg Figure 1, 2, 4), part of Figure 7 moved off the page. Therefore, I recommend carefully editing the text of the article in accordance with the requirements of the template. This also applies to writing References.

Round 2

Reviewer 1 Report

Comments and Suggestions for Authors

Dear Authors,
thank you for improving your paper and incorporating the reviewers' comments. I have no further comments.
Best regards.